



# Underway seawater and atmospheric measurements of volatile organic compounds in the Southern Ocean

Charel Wohl[1,2,3], Ian Brown[1], Vassilis Kitidis[1], Anna E. Jones[3], William T. Sturges[2], Philip D. Nightingale[1,2,4], Mingxi Yang[1]

[1]Plymouth Marine Laboratory, Plymouth, PL1 3DH, United Kingdom
[2]Centre for Ocean and Atmospheric Sciences, School of Environmental Sciences, University of East Anglia, Norwich NR4 7TJ, United Kingdom
[3]British Antarctic Survey, Cambridge, High Cross, Madingley Road, CB3 0ET, United Kingdom
[4]Sustainable Agriculture Systems, Rothamsted Research, North Wyke, Devon, EX20 2SB, UK

*Correspondence to*: Mingxi Yang (miya@pml.ac.uk)

**Abstract.** Dimethyl sulfide and volatile organic compounds and are important for atmospheric chemistry. The oceanic emissions of biogenically derived gases, including dimethyl sulfide and especially isoprene, are not well constrained. The role of the ocean in the global budgets of atmospheric methanol, acetone and acetaldehyde is even more poorly known. In order to quantify the air-sea fluxes of these gases we measured their seawater concentrations and air mixing ratios in the Atlantic sector

of the Southern Ocean, along a ~11000 km long transect at approximately 60° S in Feb-Apr 2019. Concentrations, oceanic saturations and estimated fluxes of several simultaneously sampled volatile organic compounds (methanol, acetone, acetaldehyde, dimethyl sulfide and isoprene) are presented here. Campaign mean (± 1σ) surface water concentrations of dimethyl sulfide, isoprene, methanol, acetone and acetaldehyde were 2.60 (± 3.94), 0.0133 (± 0.0063), 67 (± 35), 5.5 (± 2.5) and 2.6 (± 2.7) nmol dm$^{-3}$ respectively. In this dataset, seawater isoprene and methanol concentrations correlated positively.

Furthermore, seawater acetone, methanol and isoprene concentrations were found to correlate negatively with the fugacity of carbon dioxide, possibly due to a common biological origin. Campaign mean (± 1σ) air mixing ratios of methanol, acetone and acetaldehyde were relatively low at 0.17 (± 0.08), 0.081 (± 0.031) and 0.049 (± 0.040) ppbv. We observed diel changes in averaged acetaldehyde concentrations in seawater and ambient air (and to a lesser degree also for acetone and isoprene), which suggest light-driven productions. Campaign mean (± 1σ) fluxes of 4.3 (± 7.4) μmol m$^{-2}$ d$^{-1}$ DMS and 0.028 (± 0.021) μmol m$^{-2}$ d$^{-1}$ isoprene are determined where a positive flux indicates from the ocean to the atmosphere. Methanol was largely

undersaturated in the surface ocean with a mean (± 1σ) net flux of -2.4 (± 4.7) μmol m$^{-2}$ d$^{-1}$, but also had a few occasional episodes of outgassing This section of the Southern Ocean was found to be both a source and a sink for acetone and acetaldehyde this time of the year, depending on location, resulting in a mean flux of -0.55 (± 1.15) μmol m$^{-2}$ d$^{-1}$ for acetone and -0.28 (± 1.22) μmol m$^{-2}$ d$^{-1}$ for acetaldehyde. The data collected here will be important for constraining the oceanic

source/sink of these gases and potentially help to elucidate the presence of common sources/sinks for these compounds.





# 1 Introduction

Dimethyl sulfide is a key source of secondary organic aerosol in the global atmosphere, likely influencing cloud formation and the albedo of the planet (Charlson et al., 1987; Lana et al., 2011). Isoprene is particularly relevant for studies of atmospheric chemistry due to its extremely fast reaction with OH (Medeiros et al., 2018). Additionally, isoprene might also

contribute to particle formation in the marine atmosphere (Arnold et al., 2009; Claeys, 2004). Oxygenated volatile organic compounds (OVOCs), such as methanol, acetone and acetaldehyde, are present ubiquitously throughout the atmosphere (Heald et al., 2008). Methanol, acetone and acetaldehyde are important for the oxidative capacity of the remote marine atmosphere (Lewis et al., 2005) and are suspected to play a role in particle formation and growth (Blando and Turpin, 2000). Acetone and acetaldehyde can react with $NO_x$ to produce the pollutant peroxyacetyl nitrate (PAN) (Atkinson, 2000). PAN can decompose

over the ocean and represent a source of $NO_x$ to the remote marine atmosphere, potentially leading to ozone production (Lee et al., 2012).

The role of the oceans in the global budget of these volatile organic compounds (VOCs) is unclear. Using the latest global climatology of DMS, the global ocean is estimated to emit about 28 (Lana et al., 2011) to 20 Tg sulfur $yr^{-1}$ (Land et al., 2014). The difference between these two estimates is mainly due to the use of different gas transfer velocity parameterisations. Lana

et al. (2011) suggest that uncertainty in the distribution of seawater DMS concentration contributes to at least as much uncertainty to the global flux as the uncertainty in the gas transfer velocity. Further in situ concentration measurements, particularly in the Southern Ocean (Jarníková and Tortell, 2016), will reduce the uncertainty of this estimate. Production of DMS in seawater is relatively complex but well-studied and involves bacterial degradation of DMSP as well as direct production of DMS by phytoplankton (Dani and Loreto, 2017). Only ~10 % of the DMS in the water column is lost due to

emission to the atmosphere (Archer et al., 2002). The largest sink of DMS in seawater is bacterial consumption (Kiene and Bates, 1990).

Global oceanic isoprene emissions have been estimated to be 0.31±0.08 Tg $yr^{-1}$ using seawater concentration data (bottom-up approach) and 1.9 Tg $yr^{-1}$ using marine air mixing ratios and an atmospheric inversion model (top-down approach) (Arnold et al., 2009). Photochemical production of isoprene at the sea surface microlayer has been suggested to be a significant source of

isoprene and could partly account for this discrepancy (Ciuraru et al., 2015). However, the only direct flux measurement over the ocean to date has found no evidence for an enhanced flux under increased light levels (Kim et al., 2017). Isoprene is mainly produced in seawater by phytoplankton (Shaw et al., 2010) and the largest removal mechanism from the water column is emission to the atmosphere (Booge et al., 2018; Palmer and Shaw, 2005), probably followed by bacterial consumption (Booge et al., 2018). The lifetime of isoprene in seawater has been estimated as 7 days (Palmer and Shaw, 2005) or 10 days (Booge et

al., 2018).

Using satellite data, Stavrakou et al. (2011) suggest that methanol is both absorbed (-48 Tg $yr^{-1}$) and emitted (42.7 Tg $yr^{-1}$) by the oceans, resulting in a net sink of -5 to -13 Tg $yr^{-1}$. Earlier global budgets by Millet et al. (2008) used atmospheric measurements of methanol and have estimated that the oceans represent a net sink of -16 Tg $yr^{-1}$, with a larger oceanic source





(85 Tg yr[-1]) and sink (-101 Tg yr[-1]). However, direct flux measurements during a transatlantic transect (Yang et al., 2013a) and
in the North Atlantic (Yang et al., 2014a) have found that the flux of methanol was consistently into the ocean (Yang et al.,
2013a). Based on those Atlantic observations, a net oceanic sink of -42 Tg yr[-1] globally was extrapolated (Yang et al., 2013a),
which is much larger in magnitude than the modelling studies suggest. In a more recent budget, Müller et al. (2016) estimate
that the ocean emits 39.4 Tg yr[-1]. However, by comparing measured and modelled methanol flux, Müller et al. (2016) also find
that their ocean methanol emission of 39.4 Tg yr[-1] represents an overestimate. In seawater, methanol is thought to be
predominantly produced by phytoplankton (Mincer and Aicher, 2016) and consumed by bacteria with a lifetime of 10-26 days
(Dixon et al., 2013; Dixon and Nightingale, 2012). Methanol is a source of carbon for methylotrophic bacteria (Dixon et al.,
2011).

The most recent global budget of acetone calculates that the ocean is both the largest source (51.8 Tg yr[-1]) and the largest sink
of acetone (-59.2 Tg yr[-1]) (Brewer et al., 2017). This results in a net oceanic sink of -7.5 Tg yr[-1] globally for acetone (Brewer
et al., 2017), which represents approximately 11 % of the total acetone sink from the atmosphere. However, using eddy
covariance flux measurements over the Pacific Ocean, Marandino et al. (2005) estimate a global net oceanic sink of -42 Tg yr[-1]. Based on eddy covariance flux observations during a transatlantic transect, Yang et al. (2014c) found that the acetone flux
can be either in or out of the ocean, depending on location. This leads to highly uncertain global extrapolations as these authors
predict the ocean to be a net sink of -1 Tg yr[-1] with a propagated uncertainty of ±19 Tg yr[-1]. In the global acetone budget by
Fischer et al. (2012) and Brewer et al. (2017), the surface seawater concentration is set to a constant 15 nmol dm[-3] in the model.
In comparison, previous observations have shown that seawater acetone concentrations in the oceans range from about 2 nmol
dm[-3] (Beale et al., 2013) to up to 41 nmol dm[-3] (Tanimoto et al., 2014). The assumption of a constant seawater concentration
will lead to errors in modelled air mixing ratios and air−sea fluxes. For example, Brewer et al. (2017) highlights the importance
of surface ocean acetone concentrations for accurately predicting atmospheric mixing ratios over the Southern Hemisphere.
Acetone is thought to be produced in the oceans primarily by photochemical degradation of organic carbon (Dixon et al., 2013)
and is consumed by microbes (Dixon et al., 2013, 2014). More recently, a biological source for oceanic acetone has also been
suggested from field measurements (Schlundt et al., 2017) and laboratory phytoplankton cultures (Halsey et al., 2017). The
typical open ocean lifetimes of acetone ranges between 5 and 55 days (Dixon et al., 2013).

The ocean flux of acetaldehyde is highly uncertain. In a global budget, the ocean was modelled to be the second largest source
at 57 Tg yr[-1] (Millet et al., 2010), which represents approximately 27 % of the total source of acetaldehyde. More recently,
using an updated air−sea exchange coefficient, Wang et al. (2019) estimate the oceanic source of acetaldehyde to be 34 Tg yr[-1]. However direct flux measurements from a transatlantic transect suggest that the oceans are both a source and a sink of
acetaldehyde (Yang et al., 2014c). These authors estimate the net oceanic emission of acetaldehyde to be much lower, around
3 (propagated uncertainty ±14) Tg yr[-1] (Yang et al., 2014c). Similar to the case for acetone, the large propagated uncertainty
is because the air and water concentrations were highly variable. In the ocean, acetaldehyde is produced by photochemical
degradation of organic carbon (Dixon et al., 2013; Zhu and Kieber 2018; Kieber et al., 1990; De Bruyn et al., 2011a). A
significant light dependant biological source for acetaldehyde has been suggested from laboratory phytoplankton cultures



(Halsey et al., 2017). Bacterial consumption of acetaldehyde was found to be rapid, resulting in very short open ocean lifetimes of less than 1 day (Dixon et al., 2013; de Bruyn et al., 2017, 2013).

To the best of our knowledge, acetone, acetaldehyde and methanol seawater concentrations in the Southern Ocean have not been measured previously. Thus their air−sea fluxes and saturations in the Southern Ocean are largely unknown. The Southern Ocean is expected to play an important role in determining the air mixing ratios of these compounds in the southern hemisphere due to the low land mass and so the paucity of dominant sources such as terrestrial vegetation.

The relatively few high resolution measurements of DMS and other VOCs in seawater (Asher et al., 2011; Kameyama et al.,
2010; Royer et al., 2016; Tortell, 2005; Tran et al., 2013) indicate that these short lived gases display spatial variability on the order of tens of kilometres (Asher et al., 2011; Royer et al., 2015) and diel temporal variability . During previous campaigns, ambient air and seawater concentrations of other VOCs have rarely been measured at a high enough frequency to explore such spatial/temporal variability (Williams et al., 2004; Yang et al., 2014a, 2014c). High-resolution underway seawater concentrations enable investigators to capture hot spots that are important for estimating regional emissions (for example of
DMS, Webb et al. (2019)) and study their diurnal variability. A fast alternation between underway ambient air and seawater measurement allows the fluxes and saturations to be determined at a high resolution. Concurrent measurement of a broad range of gases also enables correlation analyses of their concentrations and identification of common sources and sinks.

Here, we present hourly averaged ambient air and seawater measurements of a suite of simultaneously measured gases (dimethyl sulfide, isoprene, acetone, acetaldehyde and methanol) from the Atlantic sector of the Southern Ocean during the
transition from late austral summer into autumn. These measurements are used to compute hourly saturations and air−sea fluxes. These observations represent a valuable dataset of a broad range of gases in a remote region.

## 2. Experimental

### 2.1 Description of the cruise

        The measurements were made during the ANDREXII cruise from 25/02 to 14/04 2019 on board of the RRS *James*
*Clark Ross* (JCR), which is part of the ORCHESTRA project (https://orchestra.ac.uk/). The vessel transited from the Falkland Islands across Drake Passage to Elephant Island near the Antarctic Peninsula. The vessel then followed a transect along a latitude of approximately 60° S eastwards past the South Orkney Islands and the South Sandwich Islands. After that, the vessel transited further east until 30° E, and then followed a return track to repeat some stations and finished in the Falkland Islands. The sampling track of the ANDREXII cruise on board JCR is shown in Figure 1 and coloured by chlorophyll a concentration
(determined from underway WET Labs WSCHL fluorometer). The underway chlorophyll a measurements determined via fluorescence are relatively uncertain due to sensor drift but have been corrected using the fluorescence measured at 5−7 m by a sensor (WET Labs ECO-AFL/FL) on the CTD rosette (Figure 2). A range of other physical and biogeochemical parameters were also measured, such as underway $fCO_2$ (Kitidis et al., 2012, 2017), sea surface temperature (SST) measured using SBE38 Sea-Bird, and sea surface salinity (sal) monitored using a SBE45 Sea-Bird Thermosalinograph. Time series of underway



salinity, sea surface temperature as well as chlorophyll a and $fCO_2$ data are shown in Figure 2. The highest $fCO_2$ values of up to 450 µatm were observed from 01/03/19 through to 03/03/19, which corresponded to sampling of upwelling waters near the Antarctic Peninsula (Amos, 2001; Takahashi et al., 2009). The highest concentrations of chlorophyll a (up to 1.2 µg dm$^{-3}$) were observed directly to the east of the South Sandwich Islands, where the ship undertook a detailed mapping of a phytoplankton bloom.

**2.2 VOC measurements**

During ANDREXII, VOCs in seawater and ambient air were measured with a Proton Transfer Reaction-Mass Spectrometer (PTR-MS, High Sensitivity Model by Ionicon). To measure seawater concentrations, a segmented flow coil equilibrator (SFCE) was used to equilibrate underway seawater with "zero air" (Wohl et al., 2019). The underway seawater inlet of the JCR is situated at approximately 5−7 m depth and set flush with the hull. The SFCE nominally sampled from the

bottom of a small (ca. 200 cm$^3$) glass vessel that was overflowing rapidly with underway seawater. In addition to the underway measurements, approximately once a day seawater sampled from the 5m Niskin bottle was measured to verify that the ship's underway seawater inlet did not affect the measured concentrations. There was no significant difference in VOC concentration sampled from the underway seawater inlet and the 5m Niskin bottle (*t*-test, *n*=35, p<0.05).

A thermometer was installed at the seawater exhaust of the SFCE to continuously measure the seawater temperature. This

revealed that when using the SFCE continuously with very cold seawater (cruise mean 1°C) and zero air cylinders mounted outside on deck, the seawater temperature in the coil only reached 18°C (despite the water bath holding the SFCE set to 25°C). This is in contrast to earlier lab measurements and an Arctic deployment, during which the water exiting the SFCE was always 20°C with the zero air cylinder housed inside of the lab. The continuously recorded temperature was used to calculate the Henry's solubility and hence the seawater concentrations for this cruise. SFCE calibrations using MilliQ water (20°C) and

cold seawater revealed that these gases fully equilibrated regardless of the initial temperature (see appendix A).

An air inlet was installed on a 40 cm pole extending forward from the railing of the walkway in front of the ship's bridge at approximately 16 m above the ocean surface. Ambient air was pumped towards the PTR-MS via a ~90 m PTFE (Polytetrafluoroethylene) air sampling tube (o.d. 9.5 mm, wall thickness 1.5 mm) using a Vacuubrand Diaphragm pump MD 4 NT at a flow rate of circa 30 dm$^3$ min$^{-1}$. The PTR-MS subsampled from this sample tube upstream of the pump. The residence

time of ambient air in the sampling tube was approximately 6 s. The blank measurements for ambient air mixing ratios were made by diverting ambient air through a custom-made Platinum-catalyst at 450°C. The high efficiency of this Pt-catalyst at oxidizing all VOCs in air to $CO_2$ has been demonstrated elsewhere (Yang and Fleming, 2019).

PTFE solenoid valves (1/8'', Takasago Fluid Systems) controlled by the PTR-MS were used to create an hourly measurement cycle of 40 min SFCE headspace (proportional to seawater concentration), 5 min ambient air scrubbed by the Pt-catalyst at

450°C (catalyst blank) and 15 min of ambient air measurements. The VOCs of interest were measured with a dwell time of 500 ms giving a total dwell time of 8s per 5 min cycle.



### 2.2.1 Calibrations

Weekly dynamic gas phase calibrations were conducted during the cruise using a certified gas calibration standard and two mass flow controllers (Apel−Riemer Environmental Inc., Miami, Florida, USA; nominal volume mixing ratio of 500 ppbv for
acetaldehyde, methanol, acetone, isoprene DMS, benzene, toluene). Calibration slopes were typically within 10 % of each other.

A lower PTR-MS drift tube voltage of 640 V was applied during this cruise compared to Wohl et al. (2019), while other PTR-MS settings were kept the same. Thus the humidity dependence of the signal and the background was slightly different from Wohl et al. (2019). Methanol and possibly acetaldehyde showed a humidity dependence for the background at 640 V. The
calibration slope for isoprene was corrected for its humidity dependence at 640 V due to fragmentation in the PTR-MS (Wohl et al., 2019). The other VOCs did not show a humidity dependence in either the slope or background. Several different types of blanks were measured in order to compute the seawater concentrations (supplementary information S1). Due to the aforementioned humidity dependence for the background of methanol and acetaldehyde, we used blanks that were measured at the same humidity as the equilibrator headspace for those two VOCs (Table 1). For compounds that do not display a humidity
dependence in the background, hourly measurement of ambient air scrubbed by the Pt-catalyst was used as a seawater blank because of its high frequency.

Two methods were used to determine the SFCE calibration slopes for seawater methanol, acetone, and acetaldehyde concentrations during this cruise: invasion and evasion. In evasion calibrations, pure solvents were dissolved by serial dilution in MilliQ water and seawater as described previously (Wohl et al., 2019). These diluted standards were measured with the
SFCE-PTR-MS system using the same procedure as for seawater samples. During invasion experiments, a known amount of certified gas standard was added to the carrier gas, which was equilibrated with essentially VOC-free MilliQ water or very deep seawater. As with previous calibrations, these measurements on board showed that the SFCE achieved essentially full (i.e. 100%) equilibration for all the VOCs measured here except for isoprene (Wohl et al., 2019). For the fully equilibrating gases, the calibration slopes are proportional to the Henry gas solubility (H). Invasion and evasion experiments represent
independent estimates of solubility of these gases at environmentally relevant concentrations. The results from these experiments agreed only if we used solubility values that are 30 % lower for acetone and 40 % lower for methanol relative to those recommended by Burkholder et al. (2015) (see appendix A). The reference of Burkholder et al. (2015) is used here since it represents a critical tabulation of the latest experimental data. The solubilities inferred by our study are within the range of previously published solubilities.
The equilibration efficiency of isoprene was determined to be 87±9 % (± 1σ) from onboard invasion experiments. This is higher than the equilibration efficiency observed during our earlier laboratory tests, probably due to the higher seawater flow (120 instead of 100 cm$^3$ min$^{-1}$) used on this cruise. We use this measured equilibration efficiency to determine seawater isoprene concentrations. The purging factors (Wohl et al., 2019) applied to fully equilibrating gases are calculated at this higher seawater flow.





### 2.2.2 Limit of detection estimates

Here we reassess the limit of detection (LOD) of our seawater and ambient air concentration measurements. This is to address the possibilities that (i) our previous estimates probably represent an overidealized case and (ii) the LOD is dependent on the PTR-MS quadrupole settings and dwell times (Yang et al., 2013b) as well as calibration slopes (Wohl et al., 2019), which may differ between deployments. The most appropriate seawater background for the ANDREXII cruise is listed in Table 1. The mean of the 5 min Pt-catalyst blank measurements was interpolated over the time series of measurements. The standard deviation of the detrended blanks (after subtracting the interpolation) was multiplied by the gas phase calibration and represents the measurement noise ($1\sigma$) in ppbv of the ambient air measurement. To calculate the measurement noise of the seawater concentration, this mixing ratio was converted to a seawater concentration in the same way as the seawater measurement (Wohl et al., 2019). The LOD was defined as $3\sigma$. The underway ambient air and seawater data was also first averaged to 5 min means, with each hourly average containing 6 continuous 5 min means of equilibrator headspace and 2 continuous 5 min means of ambient air measurements (i.e. the first minutes after automated valve switch were discarded to account for residual air in the tubing). The measurement noise derived from this analysis was divided by the square root of the number of 5 min measurements in each hourly cycle to calculate the hourly measurement noise and limit of detection listed in Table 1.

### 2.2.3 Light-driven contamination in the seawater measurement

The SFCE was installed near the starboard windows in the main lab. During the early part of the cruise, intense sunlight sometimes shined directly at the SFCE. This led to observations of extremely high headspace mixing ratios that were presumably due to photochemical production within the SFCE. This effect disappeared instantly after covering the air–water separating tee from direct sunlight. The air–water separating tee of the SFCE was thereafter covered from direct sunlight and the blinds were kept closed from 04/03/19 onwards. The effect of this light reduction measure is illustrated in (Figure 3). Hence, daytime seawater concentrations of acetaldehyde, acetone and isoprene prior to 04/03/19 were not used in further analysis. Daytime data after 04/03/19 did not show any dependence on the ship's heading, indicating that this artefact had been satisfactorily dealt with.

### 2.2.4 Filtering of Atmospheric VOC measurements

Ambient air measurements were filtered to remove the influence of ship stack contamination. Firstly, all measurements made during a sampling cycle were discarded if the concentration of benzene or toluene was above a threshold of 0.2 ppbv. This was to eliminate small scale contamination from the ventilation pipes on the foremast. Secondly, data were discarded if the relative wind speed was less than 4 m s$^{-1}$. Thirdly, only ambient air measurements with the wind coming from 10−70° either side of the bow were used for further analysis. Filtering was carried out using 1 min averaged wind measurements from a Metek sonic anemometer installed on the foremast and resulted in the removal of 55 % of the ambient air measurements.



### 2.3 Flux calculations

The saturation (%Sat) of the surface ocean relative to the atmosphere is calculated using Eq. (1).

$$\%Sat = \left(\frac{C_w}{C_a * H}\right) * 100 \tag{1}$$

Where a saturation above 100 % corresponds to oceanic outgassing. H is the dimensionless water over liquid form of the Henry solubility.

The net air−sea flux (F, positive from sea to air) is determined using the two layer model flux equation (Liss and Slater, 1974) illustrated in Eq. (2).

$$F = k * (C_w - H * C_a) \tag{2}$$

Where the gas transfer velocity (k) is defined by Eq. (3).

$$k = \frac{1}{\frac{1}{k_w} + \frac{H}{k_a}} \tag{3}$$

To calculate the airside ($k_a$) transfer velocity, we use the following parametrization derived from direct measurements of air−sea methanol transfer (Yang et al., 2013a) (Eq. (4)). This was chosen to be the $k_a$ for all VOCs of concern since we do not expect them to differ by more than ~10 %:

$$k_a = 8814\, u_* + 6810\, u_*^2 \tag{4}$$

Here the friction velocity $u_*$ is simplistically calculated using the parameterization from Johnson (2010) (Eq.(5)).

$$u_* = u_{10} * \sqrt{1.3 * 10^{-3}} \tag{5}$$

Wind speed from a Metek sonic anemometer was adjusted to 10 m height ($u_{10}$) . For isoprene, the waterside transfer velocity ($k_w$) is calculated using the parameterisation by Nightingale et al. (2000) (Eq. (6)).

$$k_w = (0.222 * u_{10}^2 + 0.333 * u_{10}) * \left(\frac{Sc_w}{Sc_{600}}\right)^{-0.5} \tag{6}$$

$Sc_w$ is the waterside Schmidt number and $Sc_{600}$ is the Schmidt number of 600. This parametrisation most likely represents an overestimation of $k_w$ for gases that have similar or greater solubility than DMS because of the solubility dependence in bubble-mediated gas exchange (Yang et al., 2011). Thus for DMS, acetaldehyde, acetone and methanol, mean $k_w$ determined from DMS measurements from five different cruises was used here (Yang et al., 2011) and scaled to the ambient temperature assuming $Sc_w^{-0.5}$.





The water phase Schmidt numbers ($Sc_w$) of methanol, acetone, acetaldehyde and DMS are determined following Johnson
(2010). The Schmidt number of isoprene is calculated using the equation presented in Palmer and Shaw (2005). Henry
solubility values are converted from freshwater to seawater using the method presented by Johnson (2010). Methanol and
acetone concentrations, fluxes and saturations are calculated using the experimentally determined solubility presented in the
appendix A.

## 3. Underway ambient air mixing ratios, seawater concentrations and air−sea fluxes

In the following sub-sections, the ambient air and seawater concentrations of DMS, isoprene, methanol, acetone and
acetaldehyde as well as their saturations and fluxes are discussed. Saturations below 100 % indicate undersaturation in seawater
(i.e. air−to−sea, or negative flux). Two versions of fluxes are presented: fluxes when both ambient air and seawater data were
available, and continuous flux estimates despite missing ambient air data (e.g. wind direction out of sector), which were
estimated by smooth interpolation of the ambient air mixing ratios.

260           As a quick overview and for reference, cruise mean air and seawater concentrations are presented in Table 2. Cruise
mean saturations and calculated fluxes are shown in Table 3. Also included are the median and quantiles as well as the standard
deviation.

### 3.1 Dimethyl sulfide

         The time series of DMS ambient air and seawater concentrations as well as the corresponding fluxes and saturations
are presented in Figure 4.

The campaign mean seawater concentration of DMS was 2.60 nmol dm$^{-3}$. The highest DMS seawater concentrations were
observed near the Antarctic Peninsula upwelling region (around 28/02/19, up to 7.55 nmol dm$^{-3}$) and east of the South
Sandwich Islands (around 13/03/19, up to 24.44 nmol dm$^{-3}$). Chlorophyll a was also elevated in those regions. These and other
fine-scale hot spots of DMS were well resolved due to our use of continuous and fast-responding measurements. To remove
the effect of ship sampling bias on the overall cruise mean (e.g. spending multiple days surveying a plankton bloom), the DMS
concentrations were first averaged in 1° longitudinal bins. The mean of spatially averaged seawater DMS concentration for
this campaign was 1.87 nmol dm$^{-3}$ (confidence interval of the mean: 1.46-2.28 nmol dm$^{-3}$), similar to the Lana et al. (2011)
climatology in this region and during these months (average of 1.5 nmol dm$^{-3}$ and range: 0−3 nmol dm$^{-3}$).

Cruise mean and median ambient air mixing ratios of DMS were 0.17 ppbv and 0.16 ppbv respectively. These values are
comparable to previous measurements over the Southern Ocean at this time of year (Bell et al., 2015; Colomb et al., 2009;
Curran et al., 1998; Guérette et al., 2019; Koga et al., 2014; Yang et al., 2011). Ambient air mixing ratios were up to about 0.5
ppbv on occasions, and do not correlate with seawater concentrations. This is probably because air parcels travel much faster
than seawater, leading to a decoupling between air and sea DMS concentrations.





The campaign mean DMS flux was 4.3 µmole m$^{-2}$ d$^{-1}$. Fluxes were typically < 7 µmole m$^{-2}$ d$^{-1}$ but exceeded 30 µmole m$^{-2}$ d$^{-1}$

within the phytoplankton bloom encountered on around 13/03/2019. Our mean DMS flux compares well to direct measurements of DMS flux over the Southern Ocean (Bell et al., 2015; Yang et al., 2011).

### 3.2 Isoprene

The time series of isoprene ambient air and seawater concentrations as well as the corresponding fluxes and saturations are presented in Figure 5.

The campaign mean isoprene seawater concentration was 0.0133 nmol dm$^{-3}$. This is comparable to previous measurements in the open ocean (Hackenberg et al., 2017; Ooki et al., 2015) and also in the Southern Ocean (Kameyama et al., 2014). Isoprene concentrations as high as 0.040 nmol dm$^{-3}$ were observed near the Antarctic Peninsula and in the phytoplankton bloom near the South Sandwich Islands. As shown in Fig. 2 and Fig. 3, these areas were also associated with higher chlorophyll a concentration and low fCO$_2$.

The linear regression between underway isoprene (nmol dm$^{-3}$) and chlorophyll a (µg dm$^{-3}$) yielded a slope of 0.0136 nmol dm$^{-3}$ isoprene (µg chla dm$^{-3}$)$^{-1}$ with an R$^2$ value of 0.35 and an intercept of 0.0087 nmol dm$^{-3}$ isoprene (P= 0.000, N=799). There also appears to be a first order relationship between chlorophyll and seawater isoprene concentrations in other oceanic basins, with variable R$^2$ values of 37% (Kameyama et al., 2014), 12 % (Baker et al., 2000) and 52 % (Broadgate et al., 1997). The regression slope from our campaign, where SST was generally between 0 and 2°C, compares best to previous measurements

in colder waters. For example, Ooki et al. (2015) have found a slope of 0.0143 nmol dm$^{-3}$ isoprene (µg chla dm$^{-3}$)$^{-1}$ and intercept of 0.00223 nmol dm$^{-3}$ isoprene in waters with temperatures between 3.3−17°C. Hackenberg et al. (2017) have found slopes of 0.0379 nmol dm$^{-3}$ isoprene (µg chla dm$^{-3}$)$^{-1}$ and 0.0341 nmol dm$^{-3}$ isoprene (µg chla dm$^{-3}$)$^{-1}$ for SST below 20°C in the Atlantic and Arctic Oceans respectively. The slope between chlorophyll a vs. isoprene concentration appears to increase in steepness with temperature (Hackenberg et al., 2017; Ooki et al., 2015).

Our dataset showed a significant negative correlation between isoprene and fCO$_2$ (slope: -0.00013 nmol dm$^{-3}$ isoprene (µatm fCO$_2$)$^{-1}$, intercept: 0.0589 nmol dm$^{-3}$ isoprene, R$^2$ 0.33, P=0.000, N=690). This might be because isoprene is produced by phytoplankton (Dani and Loreto, 2017; Shaw et al., 2010), and high biological productivity tends to reduce seawater fCO$_2$ in phytoplankton blooms (Blain et al., 2007; Wingenter et al., 2004). A negative correlation between the temperature-normalized partial pressure of CO$_2$ (pCO2) and seawater isoprene concentrations has been reported previously (Kameyama et al., 2014)

but this correlation only held for waters south of 53° S. In the study of Kameyama et al. (2014), the SST normalised pCO$_2$ was viewed as a proxy for net community production. In the same study (Kameyama et al., 2014), the correlation between isoprene and fCO2 was found to hold without this temperature normalisation, although with a reduced level of significance.

The mean ambient air mixing ratio of isoprene on this cruise was 0.053 ppbv and the median was 0.045 ppbv, illustrating a positive skewness in the isoprene ambient air mixing ratio. This positive skewedness is probably caused by biology- and wind

speed-dependent emissions as well as the short lifetime of isoprene in the atmosphere that prevents it from being more fully mixed. Positively skewed atmospheric isoprene mixing ratios have also been observed previously over the ocean (Kim et al.,



2017). The mean of our measurements compares best to previous measurements over the Southern Ocean (Colomb et al., 2009; Nadzir et al., 2019; Yokouchi et al., 1999) as well as other biologically productive areas (Shaw et al., 2010).

Isoprene was supersaturated by 760 % in the mean, leading to a mean flux of 0.028 µmole m$^{-2}$ d$^{-1}$. This flux exceeded 0.07

µmole m$^{-2}$ d$^{-1}$ on occasions. Our fluxes compare well to some published estimates from other oceans (Baker et al., 2000; Tran et al., 2013), but they are about 10-fold lower than an estimate from the Southern Ocean by Kameyama et al. (2014). This is probably due to the lower seawater concentrations measured during our campaign compared to the seawater concentrations reported by Kameyama et al. (2014). Our fluxes are also comparable to direct flux measurements in the Labrador Sea where mean isoprene fluxes were found to be dominated by episodic emissions (Kim et al., 2017).

**3.3 Methanol**

The time series of methanol ambient air and seawater concentrations as well as the corresponding fluxes and saturations are presented in Figure 6.

Median and mean seawater methanol concentrations were the same at 67 nmol dm$^{-3}$. The mean of these seawater concentrations is within previously published measurements of 15 to 361 nmol dm$^{-3}$ (Beale et al., 2013, 2015; Kameyama et al., 2009;

Williams et al., 2004; Yang et al., 2013a, 2014a). Measurements using laboratory phytoplankton cultures suggest that methanol may be produced by a broad range of phytoplankton (Mincer and Aicher, 2016). Regression analysis of seawater concentrations of methanol against isoprene gave a significant positive relationship (slope: 3524 nmol dm$^{-3}$ methanol (nmol dm$^{-3}$ isoprene)$^{-1}$, intercept: 22 nmol dm$^{-3}$ methanol, R$^2$ =0.38 P=0.000, N=771). However, seawater methanol concentrations did not correlate significantly with chlorophyll a. The correlation between methanol and isoprene suggests that both compounds may be

produced by similar phytoplankton species. Measurements of laboratory phytoplankton cultures show that cyanobacteria (*Synechococcus* and *Trichodesmium*) are strong producers of isoprene (Bonsang et al., 2010), but weak producers of methanol (Mincer and Aicher, 2016). In contrast, *Phaeodactilum*, a temperate diatom, was found to produce large amounts of methanol (Mincer and Aicher, 2016) but moderate amounts of isoprene (Bonsang et al., 2010). *Emiliania Huxley*, a coccolithophore, was observed to produce moderate amounts of both isoprene and methanol (Bonsang et al., 2010; Mincer and Aicher, 2016).

On this cruise, methanol significantly correlated with fCO$_2$ (slope: -0.97 nmol dm$^{-3}$ methanol (µatm fCO$_2$)$^{-1}$, intercept: 422 nmol dm$^{-3}$ methanol, R$^2$ 0.55, P=0.000, N=651), which is qualitatively consistent with production of methanol by phytoplankton. Unfortunately no plankton composition measurements were made during our cruise so we are unable to comment further.

Ambient air mixing ratios of methanol were very low (mean= 0.17 ppbv, median= 0.17 ppbv), in agreement with previous

measurements in the Southern Hemisphere of about 0.2 ppbv in the South Atlantic (Yang et al., 2013) and up to 0.54 ppbv above the Southern Indian Ocean (Colomb et al., 2009). Lower ambient air mixing ratios of methanol in the Southern Hemisphere compared to the Northern Hemisphere are probably due to the relatively sparse landmass and vegetation coverage (Yang et al., 2013).





The average methanol flux was calculated to be into the ocean with a mean saturation of 83 % and flux of -2.4 µmol m$^{-2}$ d$^{-1}$.

Highest seawater concentrations of up to 226 nmol dm$^{-3}$ of methanol were observed in the phytoplankton bloom encountered around 13/03/19. This is higher than previous high latitude measurements in the South Atlantic in the austral spring (Beale et al., 2013; Yang et al., 2014c) and in the Labrador sea in late boreal autumn (Yang et al., 2014b) but are similar in magnitude to measurements in parts of the North Atlantic (Beale et al., 2013). The presence of waters with high methanol concentrations, combined with relatively low ambient air mixing ratios, led to episodes of outgassing of methanol (up to ~10 µmol m$^{-2}$ d$^{-1}$).

Net sea−to−air transfer of methanol is somewhat unexpected given the extremely high solubility of methanol. Previous direct flux measurements of methanol along a meridional transect through the Atlantic (Yang et al., 2013a) and in the Labrador sea (Yang et al., 2014b) have shown that the flux of methanol was consistently into the ocean, with the largest air-to-sea flux in regions downwind of continents. Outgassing of methanol from the ocean has been suggested previously for some waters of the North Atlantic (Beale et al., 2013). In our calculation, we note that the methanol flux is insensitive to the choice of

solubility. If we instead calculated the methanol flux and seawater methanol concentrations using the recommended solubility by Burkholder et al. (2015), the mean seawater concentration of methanol, saturation and flux would have been 125 nmol dm$^{-3}$ (26 % higher) 83 % (unchanged), -2.4 µmol m$^{-2}$ d$^{-1}$ (unchanged) respectively. Saturation and flux remain unchanged since seawater concentration and solubility change by the same factor, which cancels out.

### 3.4 Acetone

The time series of acetone ambient air and seawater concentrations as well as the corresponding fluxes and saturations are presented in Figure 7.

The mean (± 1σ) seawater acetone concentration of 5.5 ±2.5 nmol dm$^{-3}$ compares well to previous measurements of less than 10 nmol dm$^{-3}$ in the South Atlantic (Beale et al., 2013; Yang et al., 2014c) and in the Labrador sea (Yang et al., 2014a). Seawater acetone concentrations from this cruise are also similar to other open ocean measurements (Hudson et al., 2007;

Kameyama et al., 2010; Marandino et al., 2005; Schlundt et al., 2017). The median acetone concentration here was 5.1 nmol dm$^{-3}$ and hence close to the campaign mean. A significant negative correlation of acetone with fCO$_2$ is observed (slope: -0.0469 nmol dm$^{-3}$ acetone (µatm CO$_2$)$^{-1}$, intercept: 22 nmol dm$^{-3}$ acetone, R$^2$ 0.55, P=0.000, N=671), which excludes high seawater acetone measurements from 08/04/19 and 10/04/19. These elevated data are considered strong outliers for reasons currently unknown, as values are higher than the upper quantile plus three times the interquantile range. This correlation of

acetone with fCO$_2$ suggests a possible role for biology in the production of acetone. Previous investigators have found correlations between seawater acetone concentration and the abundance of haptophytes and pelagophytes (Schlundt et al., 2017), suggesting direct production by phytoplankton and/or bacterial communities associated with these phytoplankton. Taddei et al. (2009) have also observed higher emission of acetone in high chlorophyll a areas in the remote South Atlantic. Our acetone data showed a weak, although significant positive correlation with chlorophyll a concentration (slope: 4.84 nmol

dm$^{-3}$ acetone (µg chla)$^{-1}$, intercept: 4.11 nmol dm$^{-3}$ acetone, R$^2$ 0.07, P=0.000, N=750). Despite this, the main source of acetone in seawater is probably photochemical production, which has been found to account for up to 100 % of gross production rates





of acetone in seawater (Dixon et al., 2013). The underway acetone air and water concentrations presented here show a small but statistically significant difference between daytime and nighttime, which will be discussed further in section 4.

The mean (± 1σ) ambient air mixing ratio of acetone measured during this cruise was very low (mean of 0.081 ±0.031 ppbv

and median 0.076 ppbv). This compares well with clean marine air measurements of 0.188 ppbv at Cape Grim, Tasmania (Galbally et al., 2007), air coming off Antarctica with an average of 0.128 ppbv (Legrand et al., 2012) and marine air measurements with an average of 0.127 ppbv over the South Atlantic at 55° S (Williams et al., 2010). The mean ambient air mixing ratio reported here is considerably lower than the modelled annual mean acetone air mixing ratio over the Southern Ocean of 0.2 about ppbv (Fischer et al., 2012). An updated global budget of acetone predicts slightly lower annual mean air

mixing ratios over the Southern Ocean of 0.1-0.2 ppbv (Brewer et al., 2017). This decrease is largely due to an increased photolysis rate of acetone in the updated model (Brewer et al., 2017). Both of these works assume a fixed acetone seawater concentration of 15 nmol dm$^{-3}$ (nearly three times higher than our measurements), and so have the potential to overestimate air mixing ratios above the Southern Ocean. Further measurements are needed to capture the seasonality of seawater acetone. The mean seawater saturation of acetone was 88 %. Saturations of between 50 and 200 % are typical for acetone (Schlundt et

al., 2017; Yang et al., 2014a, 2014c). A mean net flux into the ocean of -0.55 µmol m$^{-2}$ d$^{-1}$ suggests that the net flux of acetone is on average into the Southern Ocean this time of the year. Though occasional outgassing was also observed. Using a $t$-test, the mean acetone flux was found to be significantly different from zero and the confidence interval of the campaign mean flux was -0.44 to -0.67 µmol m$^{-2}$ d$^{-1}$. The mean flux reported here is within the uncertainties of direct flux measurements of acetone over the Atlantic, who report an mean flux of -0.2 (propagated uncertainty 2.5) µmol m$^{-2}$ d$^{-1}$ (Yang et al., 2014c). Acetone air

mixing ratios over the North Pacific were negatively correlated with the flux of acetone, which has been measured to be consistently into the ocean (Marandino et al., 2005). The dataset here did not show a significant relationship between acetone flux and acetone air mixing ratios, probably due to the paucity of atmospheric sources of acetone in this region. In fact, no correlation could be observed in the ambient air mixing ratios 1) among all the VOCs, and 2) between atmospheric VOCs and atmospheric $CO_2$. This is contrary to observations by Yang et al. (2014c), who have found that methanol, acetone and

acetaldehyde ambient air concentrations correlated between each other and with $CO_2$. These earlier air measurements were taken along a transatlantic cruise and were more impacted by terrestrial emissions (Yang et al., 2014c). The global budget of acetone suggests that the that the Southern Ocean is a weak sink for acetone (Fischer et al., 2012), in agreement with our measurements. Brewer et al. (2017) suggest that the high latitude sink of acetone depends on the atmospheric acetone air mixing ratios, sea surface temperature and wind speed. We could only find a significant relationship between the flux of

acetone and wind speed.

If the recommended solubility of Burkholder et al. (2015) is used in our calculations, the mean acetone seawater concentration, saturation and flux become 8.0 nmol dm$^{-3}$ (45 % increase), 88 % (unchanged) and -0.59 µmol m$^{-2}$ d$^{-1}$ (unchanged). The saturation and flux remain effectively unchanged, because the mean concentration and solubility change by the same factor which cancels out.



## 3.5 Acetaldehyde

The time series of acetaldehyde ambient air and seawater concentrations as well as the corresponding fluxes and saturations are presented in Figure 8.

The cruise mean seawater concentration of acetaldehyde was 2.6 nmol dm$^{-3}$, while the median concentration was 2.5 nmol dm$^{-3}$, suggesting a normal distribution in concentrations. The seawater concentrations measured here were generally lower than 6 nmol dm$^{-3}$, which compares well to other open ocean measurements (Beale et al., 2013; Kameyama et al., 2010; Schlundt et al., 2017; Williams et al., 2004; Yang et al., 2014c; Zhu and Kieber, 2018), but is lower than measurements near the coast in the English Channel (Beale et al., 2015) and off the West Coast of Florida (Mopper and Stahovec, 1986). No seawater concentrations of acetaldehyde were reported for the first four days of the deployment because of the longer time needed for acetaldehyde to be flushed from the tubing in the SFCE compared to the other VOCs. No significant correlations between seawater acetaldehyde concentrations with fCO$_2$ or with chlorophyll a were observed, possibly due to rapid oxidation of acetaldehyde in seawater (Dixon et al., 2013) that prevents the build up of significant concentrations.

Mean ambient air mixing ratios of acetaldehyde were low at 0.049 ppbv and showed limited variability. Our measurement compares well with the previous atmospheric measurements of Legrand et al. (2012), who observed an average of 0.08 ppbv acetaldehyde in ambient air off of the Antarctic continent. Our measurement is also consistent with the interhemispheric gradient in acetaldehyde concentrations, where lower ambient air mixing ratios of acetaldehyde are generally observed in the Southern Hemisphere (Galbally et al., 2007; Guérette et al., 2019; Yang et al., 2014c). Acetaldehyde showed clear diurnal variability in both seawater and ambient air, which will be discussed in more detail in Section 4.

The mean (± 1σ) saturation of acetaldehyde was 88 (±50) % , which is within the range of previously reported acetaldehyde saturations (Schlundt et al., 2017; Yang et al., 2014c). The mean flux of acetaldehyde was -0.28 µmol m$^{-2}$ d$^{-1}$ and thus into the Southern Ocean this time of the year. Using a *t*-test, we calculated that the mean acetaldehyde net flux was significantly different from zero with a confidence interval of -0.51 to -0.25 µmol m$^{-2}$ d$^{-1}$. Our measurement is within the uncertainties of direct flux measurements across the Atlantic of 0.6 (propagated uncertainty 2.5) µmol m$^{-2}$ d$^{-1}$ (Yang et al., 2014c), but is less than the estimated flux over South China and Sulu Sea at -10.11 µmol m$^{-2}$ d$^{-1}$ (Schlundt et al., 2017), probably due to the higher ambient air mixing ratios at those locations. The fluxes from our works are in approximate agreement with modelled acetaldehyde fluxes in the Southern Ocean by Wang et al. (2019), who predict that the Southern Ocean is near equilibrium with respect to acetaldehyde.

## 4. Diurnal variability in VOCs

The data was analysed for a possible diurnal cycle to obtain an indication of light-driven sources and sinks for these compounds. Dixon et al. (2013) have estimated that photochemical production accounts for up to 100 % and 68 % of the gross production rates of acetone and acetaldehyde respectively in seawater. Halsey et al. (2017) have suggested a strong light-dependant biological source for acetaldehyde and a weaker source of acetone. It might therefore be expected that these VOCs





would display diurnal changes in their seawater concentrations. Zhou and Mopper (1997) and Mopper and Stahovec (1986) have reported diurnal variability in seawater acetaldehyde off the West Coast of Florida, with the highest concentrations after solar zenith. Similarly, Takeda et al. (2014) have observed diurnal variability in acetaldehyde concentrations in an enclosed

coastal area. However, Beale et al. (2013) and Yang et al. (2014c) have found no significant difference in seawater acetone and acetaldehyde concentrations between samples collected at predawn and solar noon during crossings of the open ocean of the Atlantic. In the case of isoprene, diurnal variability in seawater concentrations has not been observed previously (Booge et al., 2018; Hackenberg et al., 2017; Moore and Wang, 2006; Tran et al., 2013) despite modelling studies suggesting its existence (Gantt et al., 2009).

450        This dataset in the Atlantic sector of the Southern Ocean shows diurnal variability in acetaldehyde, and to a lesser degree in acetone and isoprene. To illustrate this, we have taken two different approaches.  First, measurements of acetaldehyde, acetone and isoprene were put into in 24 hourly bins corresponding to the local solar time (indicated as "h") and then averaged. Second, the measurements were initially normalised by the respective daily mean concentrations and then binned-averaged. This second approach reduces the impact of spikes and short-term variability on the overall bin average, as

reflected by the generally lower relative standard deviations. These results are shown in Figure 9.

        Each hourly mean shown in Figure 9 is based on between a minimum of 8 (h13−15) and a maximum of 25 (h4) hourly measurements. The normalised bin-averages of these VOCs (i.e. the second approach above) can be found in the supplementary material (Table S2). Daytime was defined as h6−18 for this analysis, which corresponds on average to the twelve hours of sunlight. Hourly mean daytime acetaldehyde seawater concentration was 2.9 nmol dm$^{-3}$, which is 26% higher

than the mean nighttime concentration of 2.3 nmol dm$^{-3}$ (t=-3.7, P=0.002). Acetaldehyde air mixing ratios were also found to be significantly different between daytime (avg: 0.061 ppbv) and nighttime (avg: 0.040 ppbv, t=-3.7, P=0.001), a change of 53%.

        Significantly different seawater acetone concentrations were also observed during daytime (avg: 6.3 nmol dm$^{-3}$) compared to nighttime (avg: 5.8 nmol dm$^{-3}$,t=-3.8, P=0.001), albeit with only a small difference of 9%. Acetone air mixing

ratios varied between on average 0.076 ppbv at night and 0.086 ppbv during the day, again a small (13%) but significant difference (t=-3.5, P=0.003). Daytime seawater isoprene concentrations (avg: 0.0143 nmol dm$^{-3}$) were significantly higher than nighttime concentrations (0.0133 nmol dm$^{-3}$ ,t=-3.3, P=0.004) by 8%. Daytime isoprene air mixing ratios (avg: 0.056 ppbv) were significantly higher than nighttime isoprene air mixing ratios (avg: 0.050 ppbv, t=-2.6, P=0.020) by 12%. The large standard deviation compared to the standard error of each hourly bin illustrates the large variability in concentrations of these

gases. The diurnal cycle becomes more obvious in the overall bin-average in our data thanks to the large number of hourly underway samples, which reduces random noise and averages out other sources of variability, as shown by the smaller standard error. Interestingly, the amplitude of the daily cycle of these gases was not found to be significantly correlated to the light intensity. This may suggest that light intensity alone is not driving the diurnal variability of these compounds. For example, De Bruyn et al (2011) have found that the origin of dissolved organic matter strongly influences photochemical production

yields. On the other side, the phytoplankton community producing these VOCs directly could also be variable.



Over the southern Indian Ocean, previous investigators have found diel changes in ambient air acetaldehyde, acetone and isoprene mixing ratios of up to a factor of 4, 10−15 % and up to a factor of 2 respectively with maxima when solar intensity was highest (Colomb et al., 2009). The amplitude of these diurnal changes compares well to the observations presented here. The remoteness of the sampling location and the paucity of other sources possibly made it easier to detect a diurnal cycle in
the ambient air mixing ratios here.

## 5. Conclusion

This paper presents underway seawater and ambient air measurements of simultaneously measured DMS, isoprene, methanol, acetone and acetaldehyde. The measurements were taken in the Atlantic sector of the Southern Ocean along a 60° S transect
during the transition from late austral summer to early autumn. To the best of our knowledge, this represents the first set of published seawater concentrations and fluxes for methanol, acetone and acetaldehyde in the Southern Ocean. Methanol seawater concentrations were higher than previous measurements in the North Atlantic (Yang et al., 2014b), while acetone and acetaldehyde seawater concentrations were generally quite low and comparable to other high latitude measurements. The atmospheric concentrations of methanol, acetone and acetaldehyde were very low, likely due to the remoteness of the sampling
location and little influence from terrestrial emissions.

This dataset contains observational evidence for statistically significant diurnal variability in seawater and ambient air concentrations of acetaldehyde, and to a lesser degree also of acetone and isoprene. Such diurnal changes in these VOC seawater concentrations in the open ocean have not been observed before. The large number of hourly measurements and remoteness of the sampling location from terrestrial and anthropogenic influences made it possible to resolve such subtle
diurnal cycles in the marine environment.

The high resolution and frequent alternation between ambient air and seawater measurements allowed us to compute the fluxes and saturations for all of these compounds at a high temporal/spatial resolution. Methanol was transferred mostly from the atmosphere to the ocean during this cruise, giving a campaign mean flux of -2.3 µmol m$^{-2}$ d$^{-1}$. However, episodes of high methanol seawater concentrations were observed, which led to somewhat unexpected occasions of methanol outgassing from
the ocean. Acetone and acetaldehyde were both absorbed and emitted by the ocean depending on location. This sector of the Southern Ocean was calculated to be a very weak sink of acetone and acetaldehyde during this period, with a mean flux of -0.55 µmol m$^{-2}$ d$^{-1}$ and -0.24 µmol m$^{-2}$ d$^{-1}$ respectively. The high resolution measurements improve the accuracy in the estimated flux since they capture the fine scale variability in the flux direction.

Simultaneous measurement of multiple compounds allowed possible common sources and sinks to be identified. For example,
seawater methanol and isoprene concentrations were found to positively correlate, possibly due to similar biological sources for these two gases. Isoprene seawater concentrations were found to negatively correlate with $fCO_2$ and with chlorophyll a, supporting a biological origin for isoprene. Seawater acetone concentrations were found to correlate negatively with $fCO_2$,





possibly pointing towards biological production of acetone in seawater. Acetaldehyde concentrations did not clearly correlate
with the other gases, possibly due to its strong photochemical production and very rapid oxidation by bacteria (Dixon et al.,
510    2013).

The VOC concentrations presented here represent a unique dataset that can be used in models to elucidate more accurately the
role of the ocean in the global cycling of methanol, acetone and acetaldehyde, as well as to further constrain the oceanic
emissions of DMS and isoprene.

**Appendix A: Suggested solubility for acetone and methanol in seawater**

As mentioned in the main text, the invasion and evasion experiments provide two independent solubility (H) estimates at
environmentally relevant concentrations in seawater. In theory the H values determined from invasion and from evasion
experiments should agree with each other. For invasion, a certified reference gas standard (Apel–Riemer Environmental Inc.,
Miami, Florida, USA; nominal volume mixing ratio of 500 ppbv for acetaldehyde, methanol, acetone, isoprene DMS, benzene,
toluene) diluted with zero air (controlled by mass flow controllers) was used. For evasion, liquid standards produced by serial
dilution of the pure compounds were used. Most conventional methods for determining solubility of these gases rely on serial
dilution of pure solvent in water (Benkelberg et al., 1995; Clayton McAuliffe, 1971; Snider and Dawson, 1985; Zhou and
Mopper, 1990), which is challenging to do reliably at environmental concentrations because of the volatility and ease of
contamination of these VOCs (Wohl et al., 2019). The three evasion calibrations for methanol, acetone and acetaldehyde
carried out during this cruise displayed a smaller variability than observed previously (Wohl et al., 2019), possibly due to the
lower number of calibrations. The invasion calibrations during this cruise were carried out using higher input mixing ratios
than previously (up to 250 ppbv), resulting in improved signal to noise ratio (Wohl et al., 2019). Invasion and evasion
calibrations for acetone were found to agree with each other only by assuming that the solubility of acetone is 30 % lower than
that recommended by Burkholder et al. (2015) (Figure A1 and Figure A2). The response in Fig. A2 is not linear due to the
addition of a large volume of standard gas to the carrier gas, which changed the total gas flow and thus the purging factor
(Wohl et al., 2019). This was accounted for in the computation of the expected equilibrator headspace mixing ratio. The
solubility recommended from our works is however within the range of other previously published solubility values and
previous laboratory calibrations of the SFCE. It is also within the uncertainty estimate by Burkholder et al. (2015).

For methanol, we used the solubility from the evasion calibration. No invasion calibration for methanol was obtained due to
the extremely high solubility of methanol. However, the agreement between acetone evasion and invasion calibrations
provided us with confidence in the serial dilution procedure, as methanol and acetone are dissolved together during the first
step of the serial dilution. Therefore, we also suggest a 40 % lower solubility for methanol than what is recommended by
Burkholder et al. (2015) (Fig. A3). This is as well within the uncertainty of the solubility value estimated by Burkholder et al.
(2015).

In Wohl et al. (2019), the recommended solubility of these compounds from the literature (Burkholder et al., 2015) was used
to calculate seawater concentrations in order to be consistent with previous observations. Our novel method of matching up
the calibrations of these gases using evasion and invasion should lead to a more accurate determination of their solubility in
seawater at environmentally relevant concentrations. Note that the choice of solubility affects the dissolved gas concentrations,
but not the saturations or fluxes in our data. This is because $C_w$ and $C_a*H$ change by the same proportion as a function of
solubility.

The invasion and evasion calibrations for acetaldehyde do not agree with each other (Fig. A4 and Fig. A5). The evasion results
were found to agree with the recommended solubility (Burkholder et al., 2015) but the invasion results do not. This could be
due to acetaldehyde hydration reactions, which affect the air-water exchange of acetaldehyde (Bell et al., 1956; Kurz and
Coburn, 1967; Yang et al., 2014c). In fact, around 60 % of the acetaldehyde in solution is thought to be present as a hydrate
(Bell et al., 1956), but only the unhydrated form is thought to be available for air–sea exchange (Yang et al., 2014c). Bell et
al. (1956) suggest a half-life of the hydration reaction of acetaldehyde between 6 and 60 seconds. Given that the residence time
in the segmented flow tube is 40 seconds (Wohl et al., 2019) it is possible that there is not enough time for complete hydration
of acetaldehyde within the SFCE. The solubility of acetaldehyde recommended by Burkholder et al. (2015) is an apparent
solubility that represents the sum of acetaldehyde and acetaldehyde hydrate. In our study, the evasion calibration is considered
a more realistic analogue of the actual seawater measurement since liquid standards represent the sum of acetaldehyde hydrate
and pure acetaldehyde. Therefore the solubility recommended by Burkholder et al. (2015), which agrees with our evasion
calibrations, was used to compute seawater acetaldehyde concentrations, fluxes and saturations.

**Data availability**

Data presented here will be available at BODC (https://www.bodc.ac.uk/).

**Author contribution**

CW carried out the measurements and calibrations on board under the supervision of MY. PN, AJ, DC and WS provided input
to the setup on board. PN and CW wrote the Collaborative Antarctic Support Scheme proposal to secure a berth on ANDREXII.
IB measured underway seawater $CO_2$ using the setup installed with VK. CW prepared the manuscript with contributions from
all co-authors.

**Competing Interest**

The authors declare that they have no conflict of interest.



**Acknowledgements**

Thanks a lot, to the users of the PTR-MS at Plymouth Marine Laboratory who shared their time on the instrument and made it possible to take the PTR-MS on this deployment. Thanks a lot to all those involved in the transport of the PTR-MS who played a key role in making these measurements possible. Many thanks to the crew and the chief scientist, Andrew Meijers
(British Antarctic Survey), for accommodating this work. Thanks also to Richard Sims (University of Calgary) and Thomas Bell (Plymouth Marine Laboratory) for providing the seawater temperature sensors.

**Financial support**

This work was supported by the Natural Environment Research Council through the EnvEast Doctoral Training Partnership [grant number NE/L002582/1]. A berth on the ANDREXII cruise was secured through the British Antarctic Survey
Collaborative Antarctic Science Scheme. The ANDREXII cruise was financed through the ORCHESTRA project (NE/N018095/1).

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



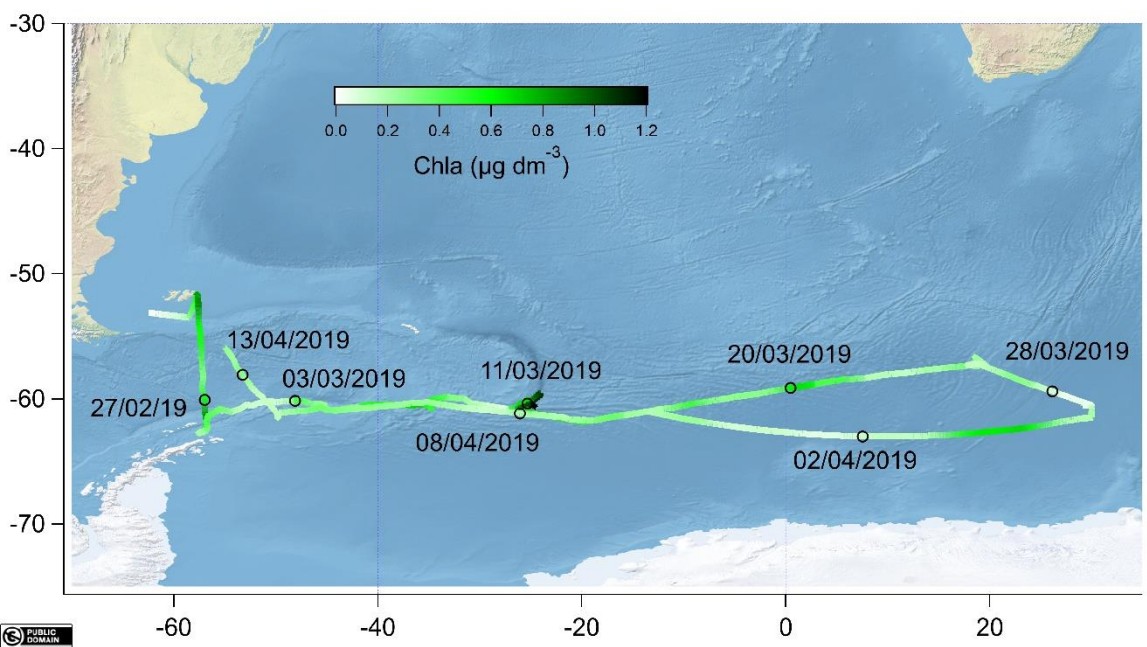

**Figure 1: Map showing the cruise track coloured by underway chlorophyll a (chla) with sampling dates indicated as black circles. All the data data was created from public domain GIS data found on the Natural Earth web site (http://www.naturalearthdata.com). It was read into Igor using the IgorGIS XOP beta.**

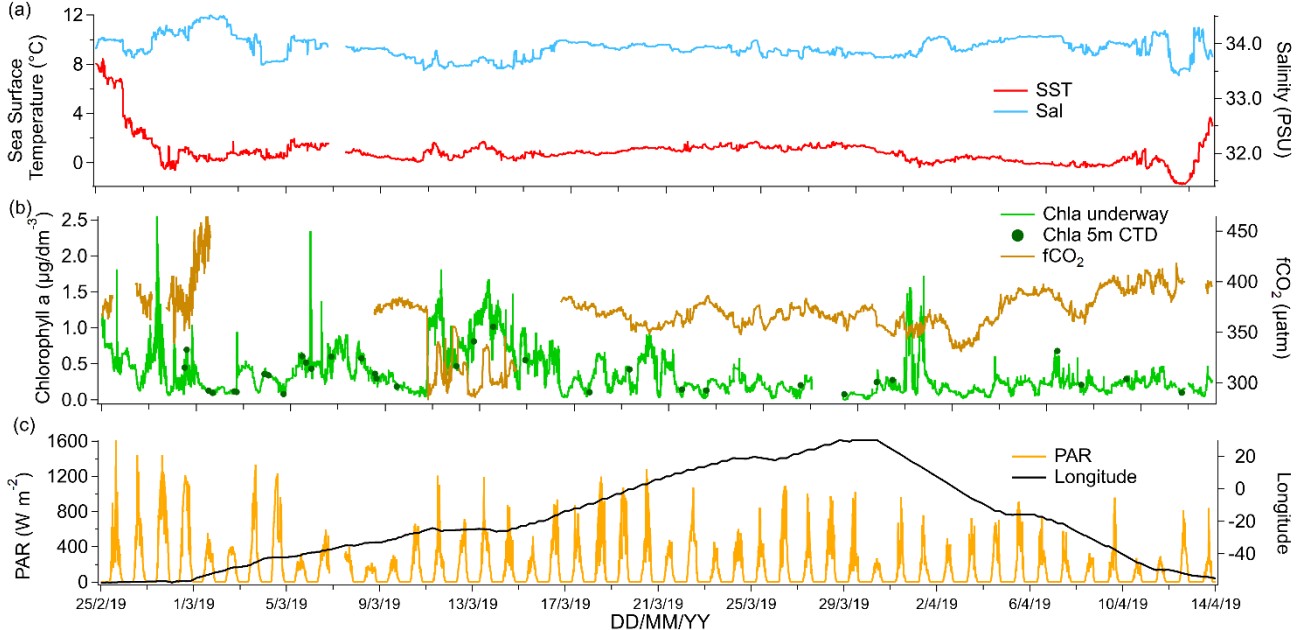

**Figure 2: (a) Sea surface temperature and surface salinity, (b) chlorophyll a concentrations measured underway and from the sensor installed on the CTD at 5 m depth as well as underway fCO₂ and (c) photosynthetic active radiation along with the longitude measured along the cruise track.**



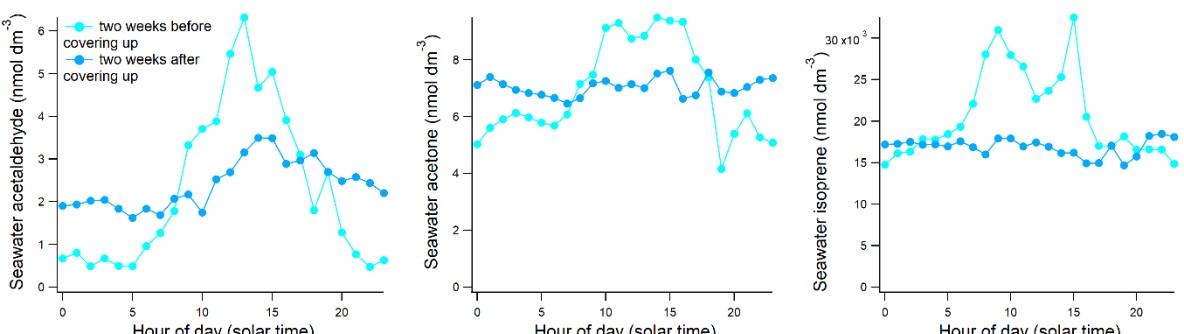

**Figure 3: Underway seawater concentrations binned in 24 hourly bins for the 2 weeks before and 2 weeks after protecting the SFCE equilibrator from sunlight on 04/03/19.**


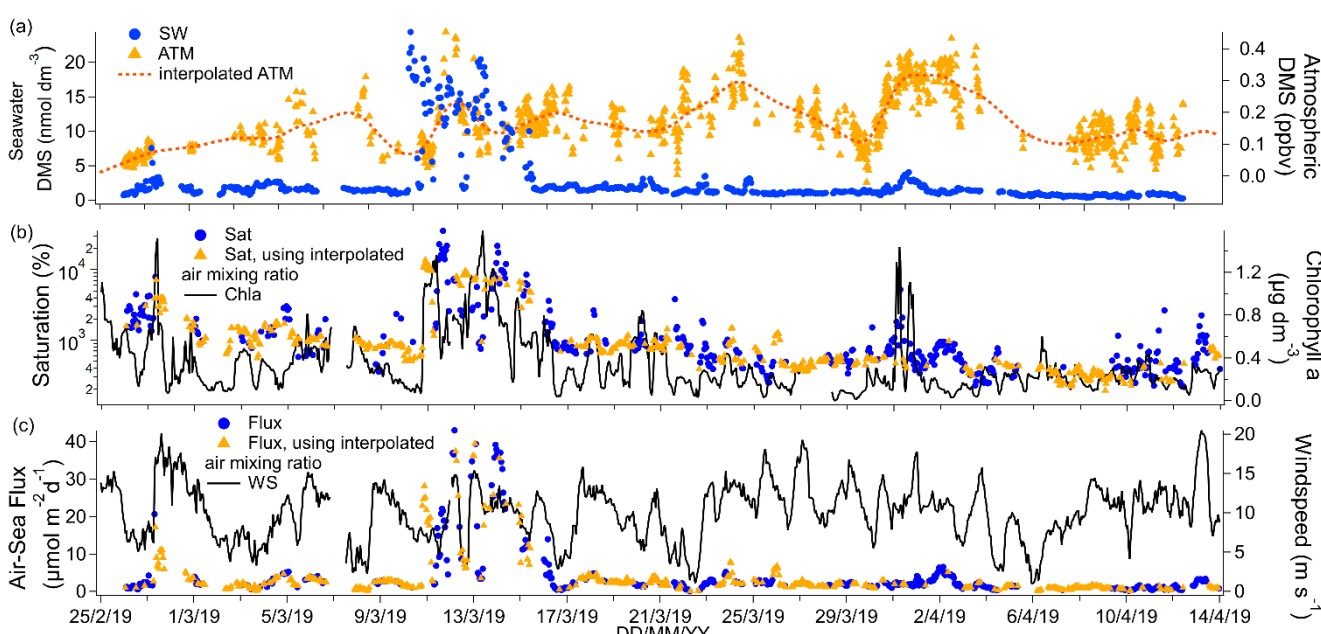

**Figure 4: (a) Time series of DMS seawater (SW) concentrations as well as measured and interpolated marine boundary layer air mixing ratios (ATM and interpolated ATM). (b) Time series of DMS saturations determined using the measured air mixing ratio and interpolated air mixing ratio and time series of chlorophyll a. (c) Time series of air−sea DMS fluxes calculated using the measured air mixing ratio and interpolated air mixing ratio and time series of wind speed.**






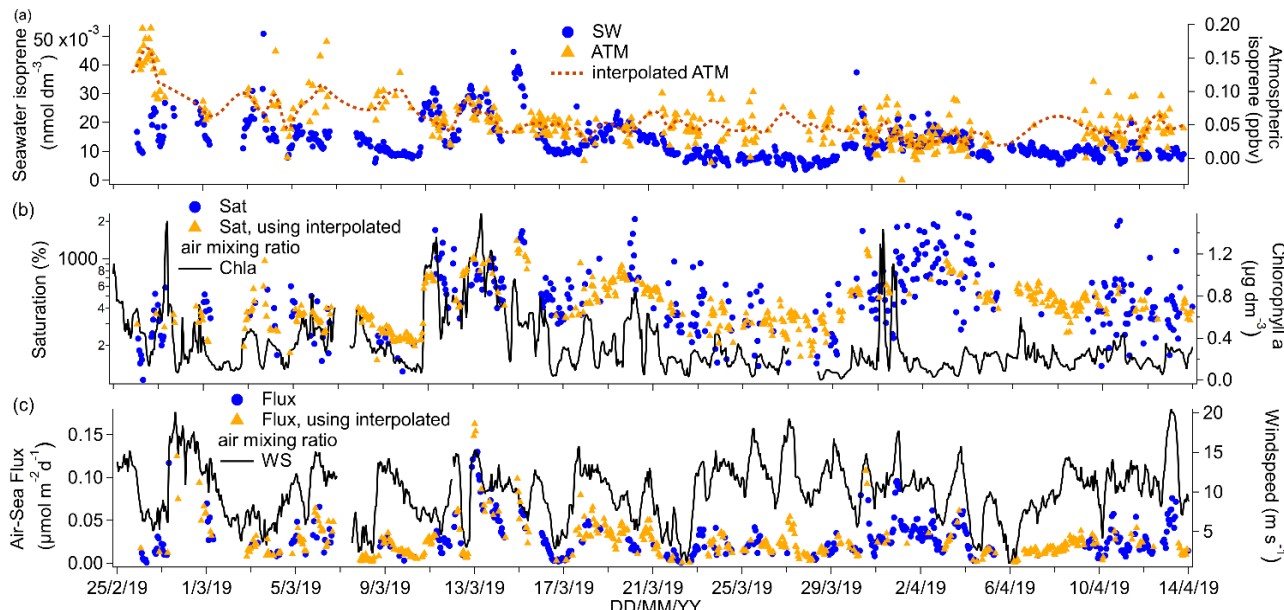

**Figure 5: (a) Time series of isoprene seawater (SW) concentrations as well as measured and interpolated marine boundary layer air mixing ratios (ATM and interpolated ATM). (b) Time series of isoprene saturations determined using the measured air mixing ratio and interpolated air mixing ratio and times eries of chlorophyll a. (c) Time series of air−sea isoprene fluxes calculated using the measured air mixing ratio and interpolated air mixing ratio and time series of wind speed.**



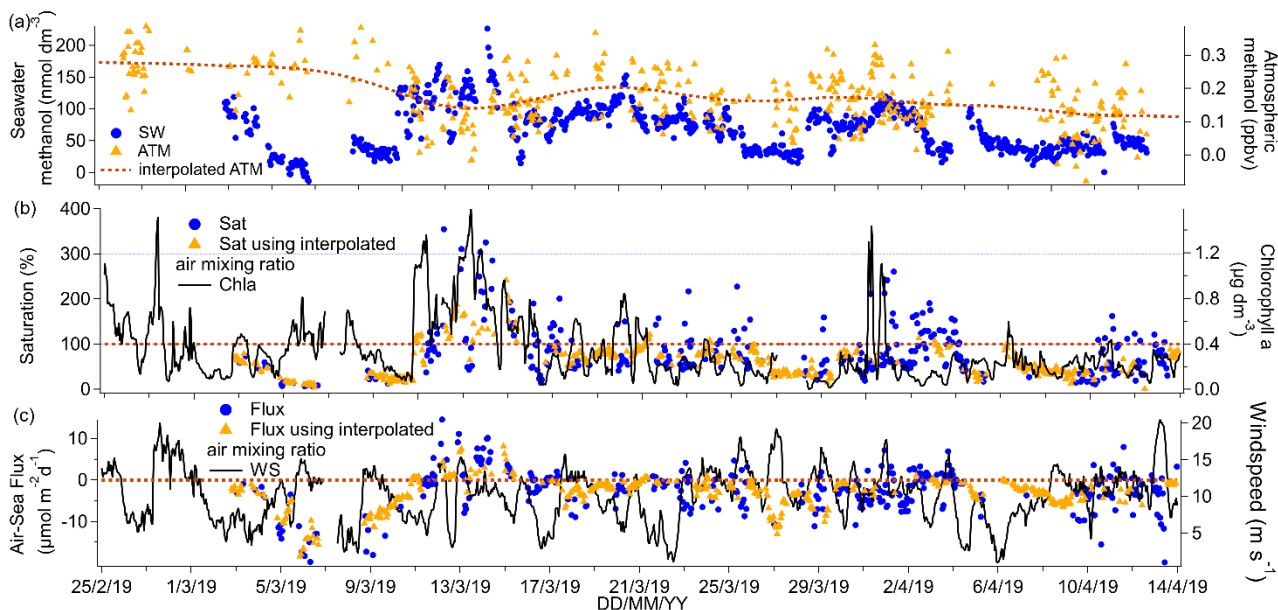

**Figure 6: (a) Time series of methanol seawater (SW) concentrations as well as measured and interpolated marine boundary layer air mixing ratios (ATM and interpolated ATM). (b) Time series of methanol saturations determined using the measured air mixing ratio and interpolated air mixing ratio and time series of chlorophyll a. (c) Time series of air−sea methanol fluxes calculated using the measured air mixing ratio and interpolated air mixing ratio and time series of wind speed.**




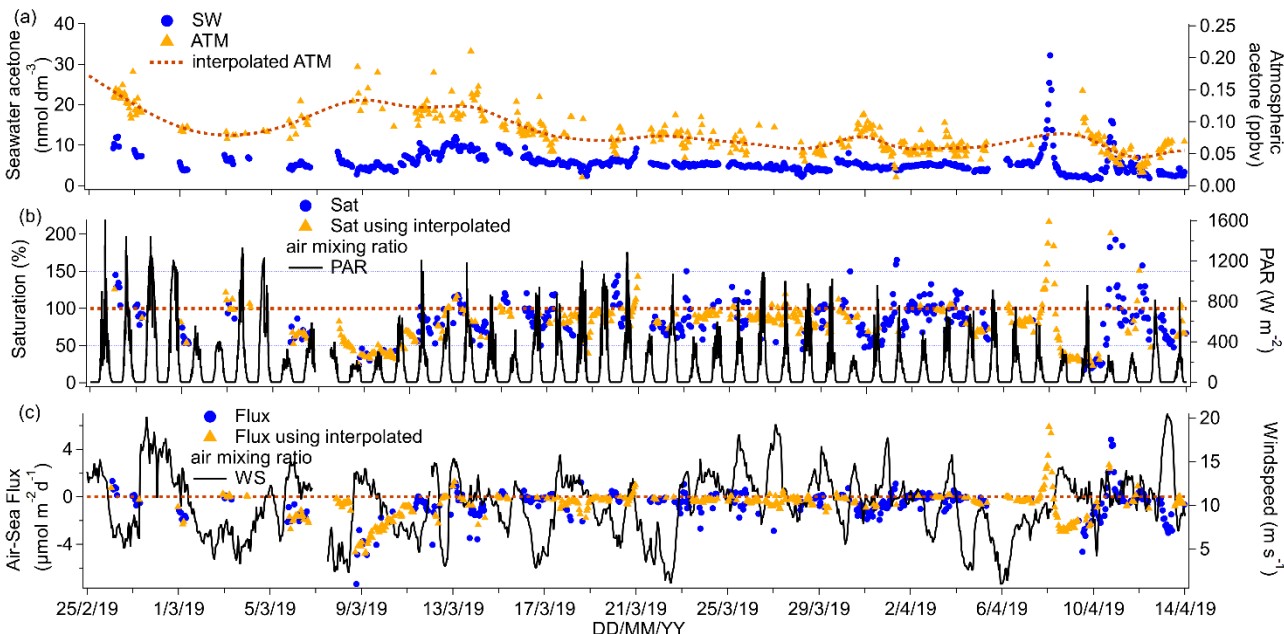

**Figure 7: (a) Time series of acetone seawater (SW) concentrations as well as measured and interpolated marine boundary layer air mixing ratios (ATM and interpolated ATM). (b) Time series of acetone saturations determined using the measured air mixing ratio and interpolated air mixing ratio and time series of chlorophyll a. (c) Time series of air−sea acetone fluxes calculated using the measured air mixing ratio and interpolated air mixing ratio and time series of wind speed.**





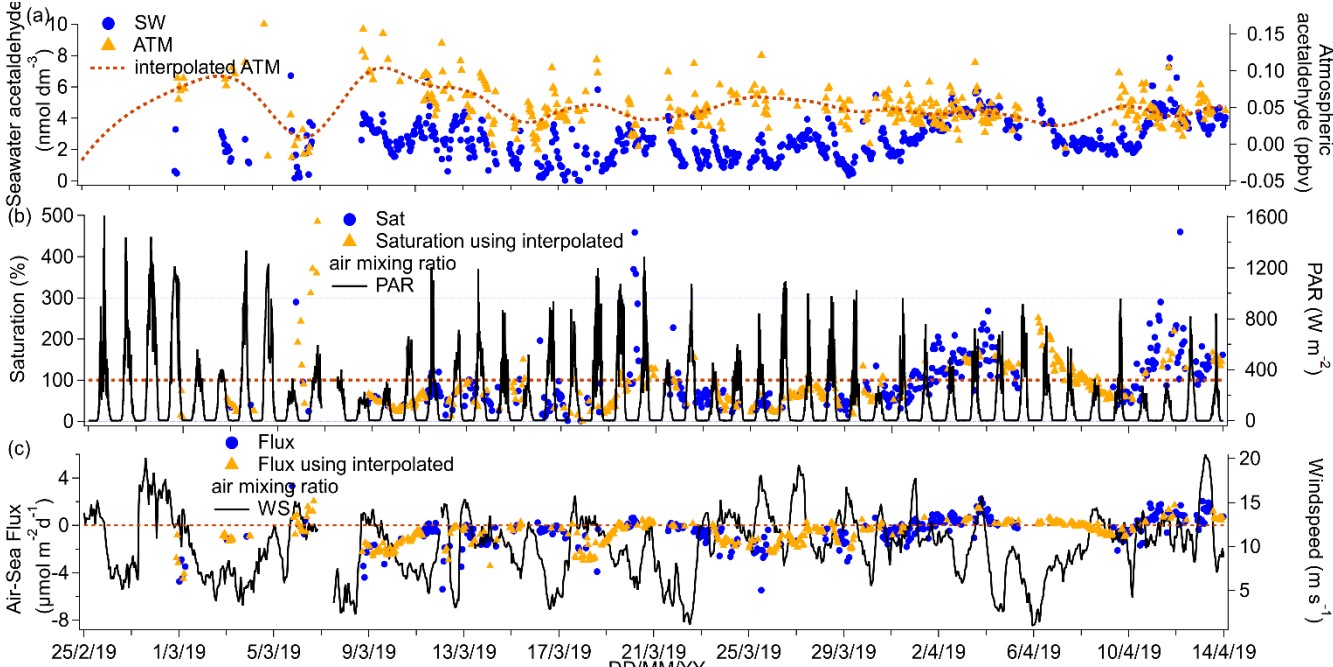

**Figure 8**: **(a) Time series of acetaldehyde seawater (SW) concentrations as well as measured and interpolated marine boundary layer air mixing ratios (ATM and interpolated ATM). (b) Time series of acetaldehyde saturations determined using the measured air mixing ratio and interpolated air mixing ratio and time series of chlorophyll a. (c) Time series of air−sea acetaldehyde fluxes calculated using the measured air mixing ratio and interpolated air mixing ratio and time series of wind speed.**





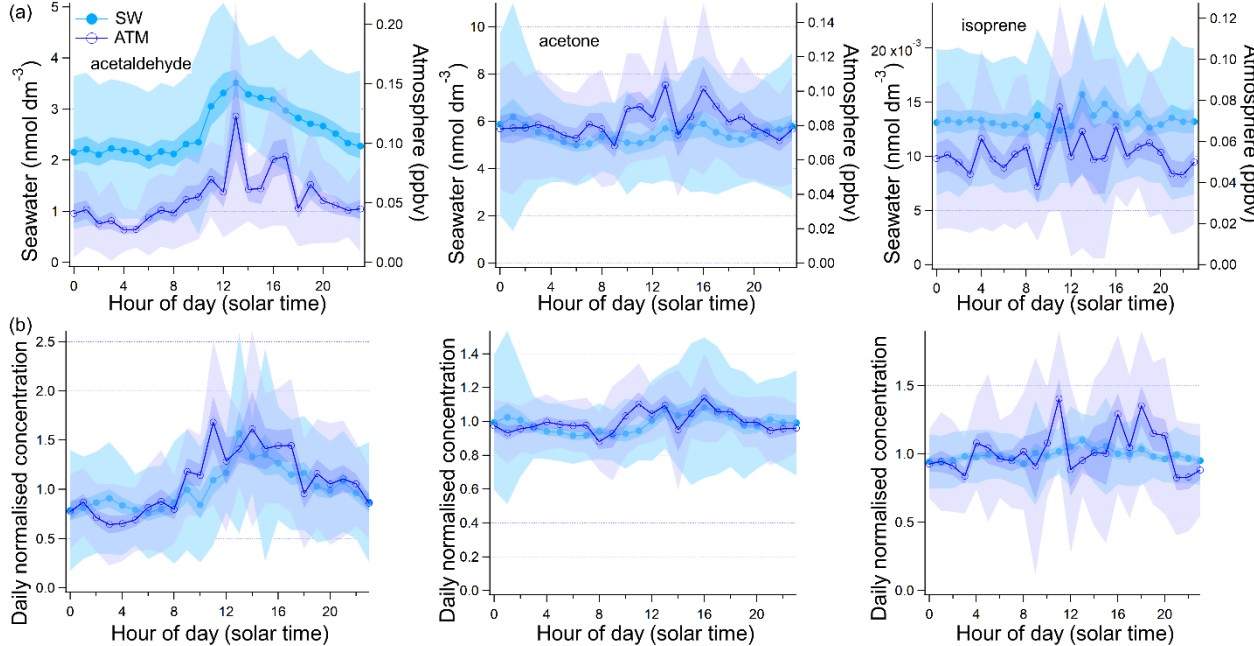

**Figure 9: Diurnal changes in seawater and atmospheric concentrations expressed (a) as true 24 hourly averaged concentration and (b) daily normalised concentration where the hourly measured concentration is divided by the average of the 24 hours that this measurement is part of. Light shaded areas show the standard deviation of each hourly bin and the darker shaded areas show the standard error of each hourly bin.**






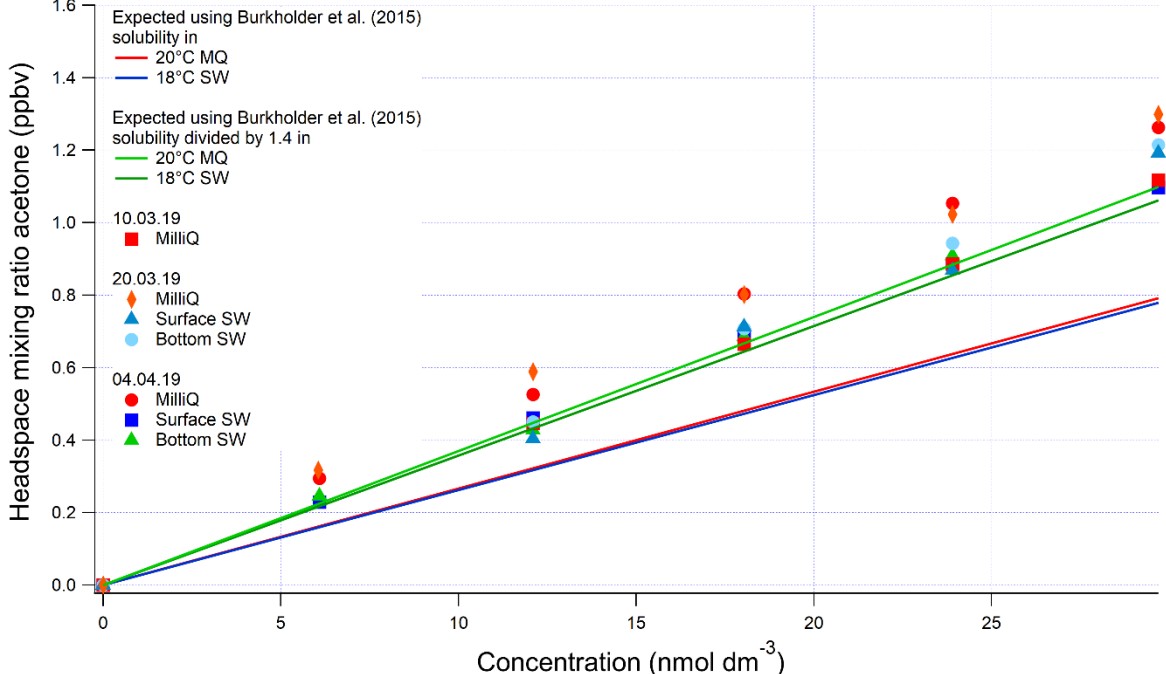

**Figure A1: Evasion calibrations using liquid standards of acetone produced by serial dilution in different types of water (SW= seawater, MilliQ= MilliQ water). Bottom SW refers to seawater collected from well below the mixed layer, near the bottom of the water column; Surface SW refers to seawater collected from the underway seawater inlet. The average measured slope in the 4 seawater calibrations is $0.0388 \pm 0.004$ (std. dev) ppbv/(nmol dm$^{-3}$) (10 % rel.std. dev.) and the average slope in the 4 MilliQ calibrations is $0.0398 \pm 0.002$ (std. dev) ppbv/(nmol dm$^{-3}$) (5 % rel. std. dev.). Expected mixing ratios are slightly different between MilliQ and SW due to the lower seawater temperature.**




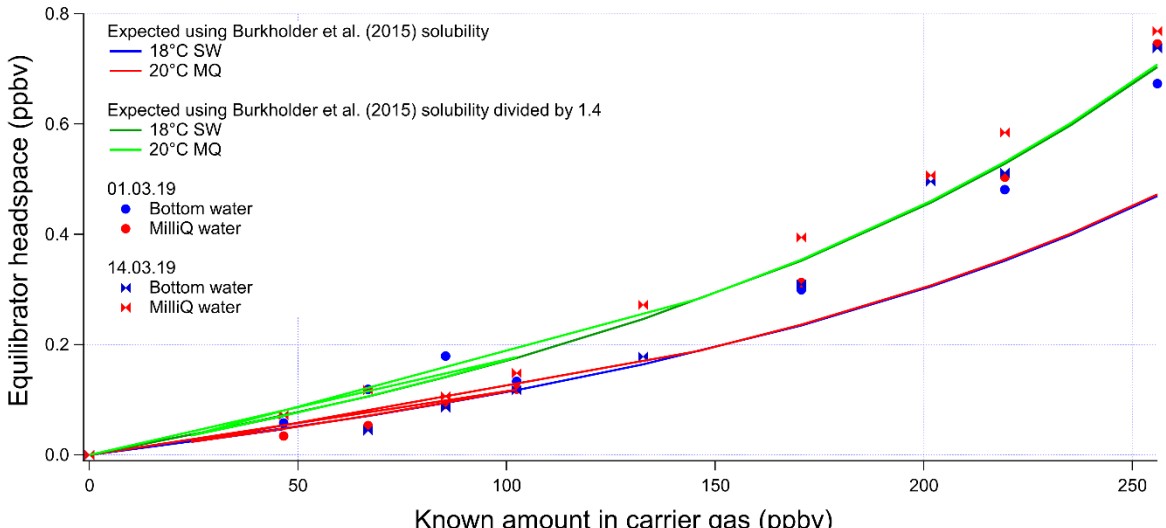

**Figure A2: Invasion calibrations for acetone carried out during the deployment and using different types of water (SW= seawater, MQ= MilliQ water). Bottom SW refers to seawater collected from well below the mixed layer. The non-linear response is due to the changing gas flow as more standard gas is added to the zero air carrier gas, which alters the purging factor.**





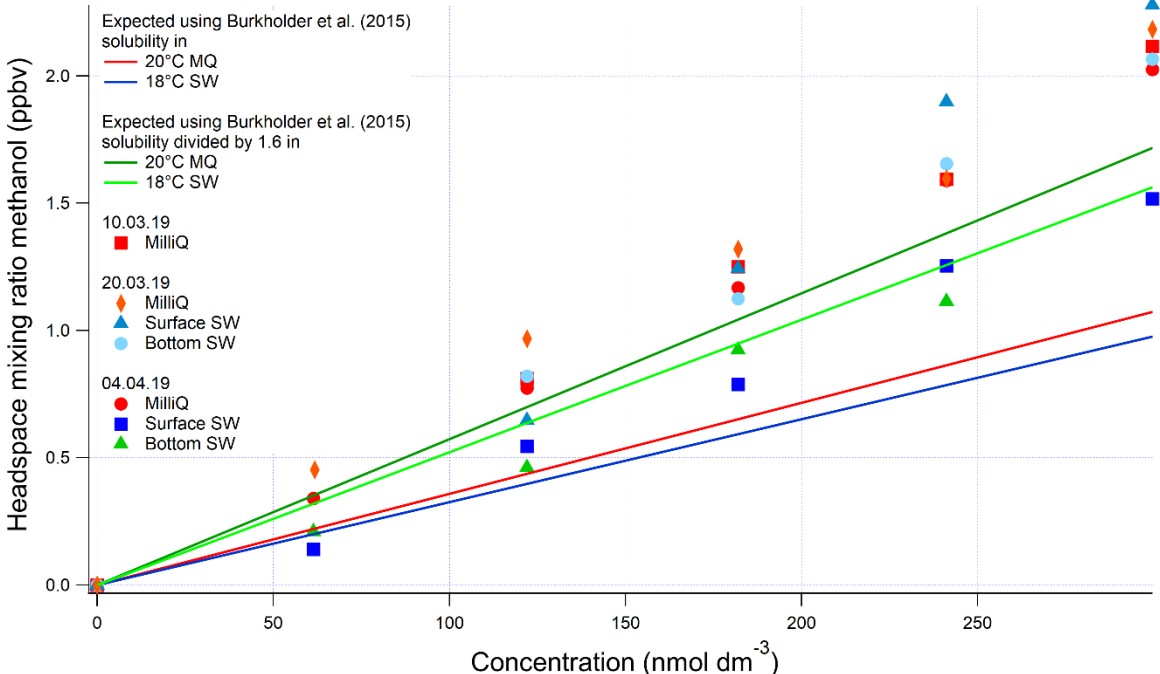

**Figure A3: Evasion calibrations using liquid standards of methanol produced by serial dilution in different types of water (SW= seawater, MQ= MilliQ water). Bottom SW refers to seawater collected from well below the mixed layer, Surface SW refers to seawater collected from the underway seawater inlet. The average measured slope in the 4 seawater calibrations is 0.00624 ± 0.00121 ppbv/(nmol dm$^{-3}$) (rel. std. dev. 19 %) and the average slope in the 4 MilliQ calibrations is 0.00678 ± 0.000254 ppbv/(nmol dm$^{-3}$) (rel. std. dev. 3 %). Expected mixing ratios are slightly different between MQ and SW due to the lower seawater temperature.**






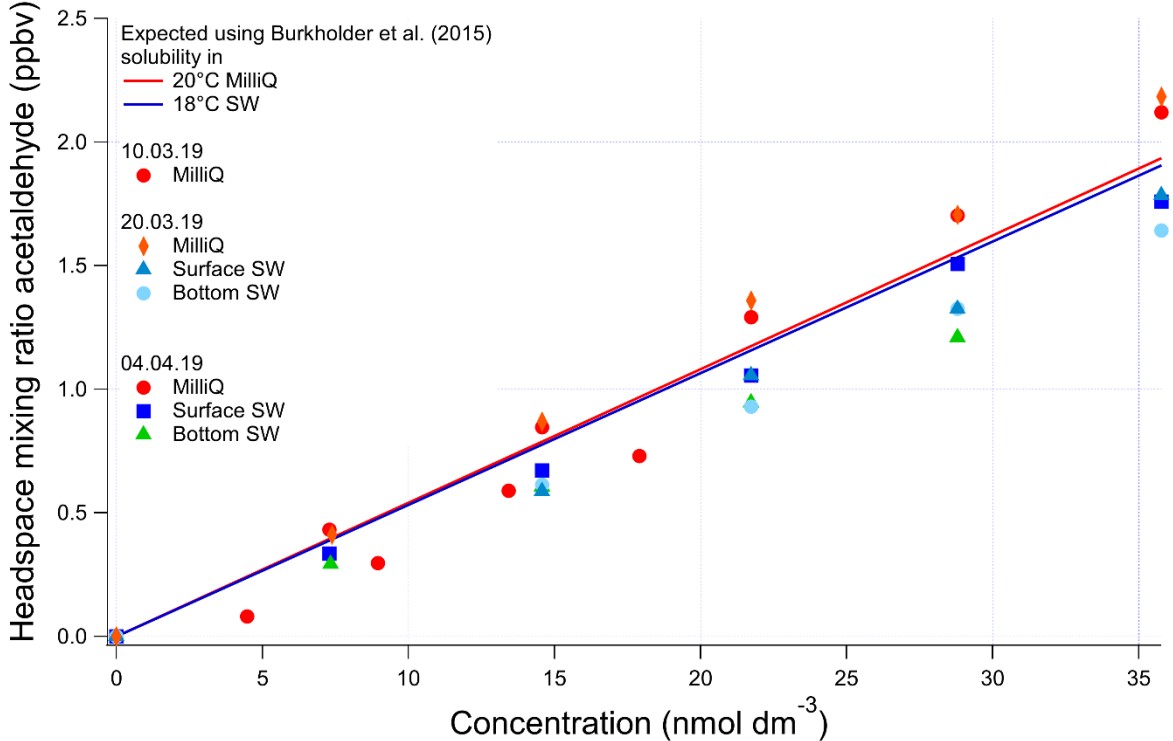

**Figure A4: Evasion calibrations using liquid standards of acetaldehyde produced by serial dilution in different types of water (SW= seawater, MilliQ= MilliQ water). Bottom SW refers to seawater collected from well below the mixed layer,, Surface SW refers to seawater collected from the ship's underway seawater inlet. The average measured slope in the 4 seawater calibrations is 0.0473 ±0.00313 ppbv/(nmol dm$^{-3}$) (rel. std. dev. 6 %) and the average slope in the 4 MilliQ calibrations is 0.0548 ±0.00767 ppbv/(nmol dm$^{-3}$) (rel. std. dev. 14 %). Expected mixing ratios are slightly different between MilliQ and SW due to the lower seawater temperature.**





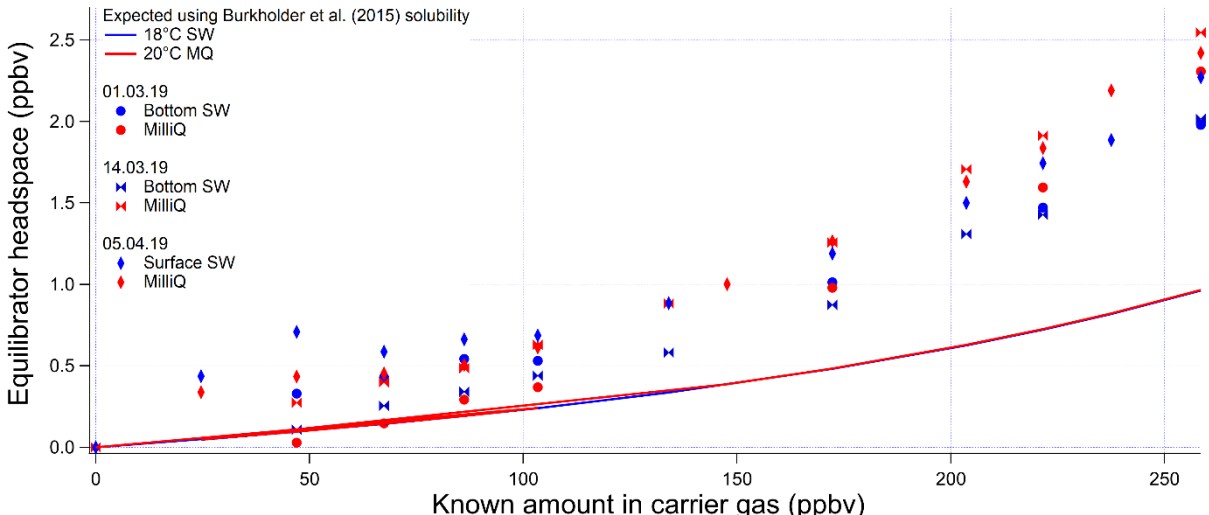

**Figure A5: Invasion calibrations for acetaldehyde carried out during the deployment and using different types of water (SW= seawater, MQ= MilliQ water). Bottom SW refers to seawater collected from well below the mixed layer, Surface SW refers to seawater collected from the ship's underway seawater inlet. The non-linear response is due to the changing gas flow as more standard gas is added to the zero air carrier gas, which alters the purging factor.**


**Table 1: Seawater blank used for each compound during this deployment is listed. Hourly measurement noise (1 σ) and limit of detection (3 σ) of seawater and ambient air measurement are listed and were determined as described in the text.**

| compound | suggested seawater blank | seawater measurement noise (nmol dm$^{-3}$) | seawater limit of detection (nmol dm$^{-3}$) | ambient air measurement noise (nmol dm$^{-3}$) | ambient air limit of detection (nmol dm$^{-3}$) |
|---|---|---|---|---|---|
| DMS | Pt-catalyst | 0.006 | 0.018 | 0.012 | 0.036 |
| isoprene | Pt-catalyst | 0.0003 | 0.0009 | 0.008 | 0.024 |
| methanol | Humid air | 7 | 21 | 0.05 | 0.15 |
| acetone | Pt-catalyst | 0.17 | 0.51 | 0.09 | 0.27 |
| acetaldehyde | Wet equilibrator | 0.4 | 1.2 | 0.014 | 0.042 |







**Table 2: Campaign mean seawater concentration (nmol dm$^{-3}$) and ambient air mixing ratio (ppbv). Campaign median and quantiles are also indicated as well as the standard deviation (s dev).**

| | | cruise mean | s dev | Q25 | median | Q75 |
|---|---|---|---|---|---|---|
| DMS | seawater | 2.6 | 3.94 | 1 | 1.39 | 1.91 |
| | ambient air | 0.17 | 0.09 | 0.10 | 0.16 | 0.23 |
| isoprene | seawater | 0.0133 | 0.0063 | 0.0089 | 0.0117 | 0.0157 |
| | ambient air | 0.053 | 0.034 | 0.031 | 0.045 | 0.065 |
| methanol | seawater | 67 | 35 | 36 | 67 | 92 |
| | ambient air | 0.17 | 0.08 | 0.11 | 0.17 | 0.23 |
| acetone | seawater | 5.5 | 2.5 | 4.3 | 5.1 | 5.9 |
| | ambient air | 0.081 | 0.031 | 0.057 | 0.076 | 0.097 |
| acetaldehyde | seawater | 2.6 | 2.7 | 1.7 | 2.5 | 3.5 |
| | ambient air | 0.049 | 0.04 | 0.025 | 0.04 | 0.061 |

**Table 3: Campaign mean saturation (%) and flux (μmol m$^{-2}$ d$^{-1}$). These calculations only include hourly fluxes and saturations for which ambient air and seawater measurements were both available. Saturations below 100 % indicate undersaturation and a negative flux indicates flux from the air into the ocean. Campaign median and quantiles are also indicated as well as the standard deviation (s dev).**

| | | cruise mean | s dev | Q25 | median | Q75 |
|---|---|---|---|---|---|---|
| DMS | sat | 1884 | 3684 | 473 | 747 | 1516 |
| | flux | 4.3 | 7.4 | 1.3 | 2.0 | 3.3 |
| isoprene | sat | 760 | 2163 | 322 | 477 | 730 |
| | flux | 0.028 | 0.021 | 0.014 | 0.023 | 0.037 |
| methanol | sat | 83 | 61 | 46 | 63 | 107 |
| | flux | -2.4 | 4.7 | -4.8 | -1.9 | 0.5 |
| acetone | sat | 88 | 41 | 69 | 84 | 101 |
| | flux | -0.55 | 1.14 | -1 | -0.29 | 0.03 |
| acetaldehyde | sat | 88 | 50 | 49 | 74 | 119 |
| | flux | -0.28 | 1.22 | -0.87 | -0.22 | 0.44 |
