# Peer review of "Underway seawater and atmospheric measurements of volatile organic compounds in the Southern Ocean"

_Biogeosciences, 2020_

## Referee Comment (RC1) · Byron Blomquist (Referee) · 2 Mar 2020

This is a valuable study and generally well-written. I have a few minor usage corrections which I'll list at the end. Here are some general comments for the authors to consider.

I wonder about sampling artifacts in the atmospheric measurements using the long inlet, described only as 90 meter, 9.5mm OD teflon tube. This inlet is probably OK for DMS but may contribute measurement bias for the more soluble species. Can the authors cite evidence to address this concern? Diverting sample flow to a Pt-catalyst combustion furnace provides an instrument blank for the PTRMS, but the authors don't mention doing a zero-air or standard injection at the inlet tip to characterize sampling

artifacts from, for example, marine aerosols accumulating inside the tubing. I'll note the inlet used by Kim (2016,2017) was much shorter, sheathed and heated for its entire length, and used impactors to limit aerosol contamination. The inlet for the Yang 2014 AMT cruise was 25m and shielded from light to prevent photochemical conversion.

Are the atmospheric concentration units (nmol/L) in Table 1 correct? Elsewhere the air values are quoted as ppbv (nmol/mol, nL/L). Many values reported in the text would fall well below the DL if nmol/L values in Table 1 are converted to ppbv. If air units on Table 1 are in fact ppbv, then project mean concentrations for all species but DMS appear to be right at the DL. I'll assume this is the case, but the authors should clarify.

There are many places in the text where the authors state their measurements 'compare well' with prior studies, but specific values from the literature are not always given. A comparison with published studies in the Southern Ocean (SO) and other regions is important but would be easier to digest if this information were removed from the various results sections, organized, and presented in Table format. A discussion of the these should be provided in a Discussion Section following the Results.

We know more about DMS than the other species in this study, and the surface ocean is unambiguously a source of DMS to the atmosphere over all seasons. Assuming the seawater concentrations and estimated fluxes observed in low-productivity areas are generally representative of fall/winter conditions over the entire SO, and the mean values from the entire cruise are typical of summer, it would be interesting to compute the estimated annual DMS emission over the entire SO region and compare this with prior estimates. Do we think the results from this cruise are representative of the SO in general? Have we now reached a reasonable consensus on annual DMS emissions from the SO?

This project is valuable because the SO is a unique marine environment, isolated from anthropogenic and continental emission sources. The cruise covered a broad swath of the SO, encountered a range of conditions relating to primary productivity, and con-

ducted the first survey of air/sea concentrations for methanol, acetone and acetalde-hyde. Readers will inevitably speculate on the broader geochemical significance of the results, so it seems to me the authors could strengthen their concluding remarks and provide their own perspective, suggesting hypotheses that emerge from this study.

For example, from what I've gathered in my brief reading: 1. The SO is supersatu-rated with isoprene, even in low productivity areas, implying a continuous source to the marine atmosphere, perhaps over all seasons. 2. Methanol, acetone (and ac-etaldehyde?) are undersaturated in the surface ocean except during episodic cases of enhanced productivity, and it is therefore likely the undersaturated condition persists throughout the fall/winter seasons when productivity is low. Thus, the SO represents a sink in the global atmospheric budgets for methanol and acetone. 3. The observed relationships to fCO2 provide a crude way to estimate localized emissions of these gas species and their impact on atmospheric oxidative capacity and aerosol produc-tion/growth. We hypothesize these atmospheric impacts are restricted to upwelling regions of high productivity.

Are these appropriate? I'd like to hear the author's thoughts.

Minor comments:

line 32: 'Dimethyl sulfide is a key source of secondary organic aerosol' - suggest you omit 'organic' since the major contribution is from inorganic sulfate, although MSA is also produced.

line 39: I wouldn't call PAN a 'pollutant' since it's a natural component of photochemical cycles in the background (unpolluted) atmosphere.

line 228: do you mean 'H is the dimensionless liquid over gas' form of H?

Finally, it seems like the Appendix and related plots on solubility could be moved to the supplemental material.

---

## Referee Comment (RC2) · Anonymous Referee #2 · 11 Mar 2020

This manuscript describes surface ocean concentrations of DMS, isoprene, acetaldehyde, acetone, and methanol and their sea-to-air flux in the Southern Ocean in late summer, early fall. This manuscript is well written and shows an interesting dataset. Especially the discussion of the measurement system and its measurements and technical issues clearly convince the reader of a robust, well analyzed dataset.

Some specific minor questions/comments are: The authors mention that, to their knowledge, these are the first reported seawater measurements for methanol, acetone and acetaldehyde. Even for DMS and isoprene the Southern Ocean is highly undersampled which, for all 5 compounds, increases errors when running global atmospheric models and using no (or very sparse) data from the Southern Ocean. This fact highlights the importance of these measurements presented in the manuscript, which I think the authors are aware of. However, I suggest to even highlight this importance in the introduction section adding a paragraph about the Southern Ocean and its influence on the atmospheric chemistry, highlighting the importance of this work. 2.1.1: Calibrations. Do the authors have any scientific explanation why isoprene, the most insoluble compound within the 5 compounds presented, is the only compound not achieving fully equilibration using the presented setup? 2.2.3: The authors mention the light driven contamination in the seawater measurements of acetone, acetaldehyde and isoprene and I am confident that they solved this issue. However, for me it is not clear what exactly causes this issue. Perhaps the authors could state clearly if they think it is coming from contamination of the material exposed to high sunlight intensity or from photochemical production in the water flowing through the tube. Both facts seem reasonable, however, if it is the material shouldn't you see these variations also when measuring outside air? Did you experience similar issues in former cruises or tests? Figure 3: Data is shown for 2 weeks before and after dealing with this issue. First, the cruise started only one week before the issue was dealt with, which leads to second, do the two subsets of data have about the same number of measurements? Please check. Additionally, the authors state that daytime values prior to 04/03/19 were not used. However, it seems, that night time data shown in Figure 3 is consistently (for all three compounds) lower than data prior to 04/03/19 shown in Fig. 5a, 7a, and 8a. (i.e. Figure 3, acetone seawater night time values "2 weeks before": ∼6 nM; Figure 7, average acetone seawater values shown prior to 04/03/19: ∼7-9nM). Please check. l. 309 / Table 2: The authors mention the positive skewness (mean: 0.053, median: 0.045) in the isoprene ambient air mixing ratio which they explain by biology and wind driven emissions as well as the very short atmospheric lifetime. I totally agree. However, this skewness is not at all discussed in the DMS section although the skewness is way higher (mean: 2.6, median: 1.39). DMS has a longer atmospheric lifetime and is more soluble than isoprene.

Technical corrections: l. 11: delete "and" after "compounds" l. 27: missing full stop after "outgassing" l. 288: "As shown in Fig. 1 and Fig. 2 . . ." l. 292: "chlorophyll a" l. 441: "dependent" l. 452: remove "in" l. 518: add "," after "isoprene" ll. 726-730: check reference typo Figure caption 1: double use of "data" Figure 2a: right y-axis: remove "(PSU)" Figure 2b: left y-axis "$\mu$g dm-3" Figure caption 5: ". . .and time series of chlorophyll a."

―――――――――――――――――――――

---

## Author Comment (AC1) · 1 Apr 2020

Many thanks for the thoughtful comments from this reviewer. The reviewer has been able to provide thought provoking comments, which clearly improved the manuscript. Please see our responses below. Reviewer comments are in italic and author's replies can be found in normal font. The changes to the manuscript are presented as figures taken from the manuscript with the changes made indicated by red track changes.

General Comments

(1) I wonder about sampling artifacts in the atmospheric measurements using the long

inlet, described only as 90 meter, 9.5mm OD teflon tube. This inlet is probably OK for DMS but may contribute measurement bias for the more soluble species. Can the authors cite evidence to address this concern? Diverting sample flow to a Pt-catalyst combustion furnace provides an instrument blank for the PTRMS, but the authors don't mention doing a zero-air or standard injection at the inlet tip to characterize sampling artifacts from, for example, marine aerosols accumulating inside the tubing. I'll note the inlet used by Kim (2016,2017) was much shorter, sheathed and heated for its entire length, and used impactors to limit aerosol contamination. The inlet for the Yang 2014 AMT cruise was 25m and shielded from light to prevent photochemical conversion.

We discuss below two aspects of potential bias related to the use of the long air sampling tube: attenuation of the gas signal (e.g. adsorption to the tube wall), contamination of the gas signal (e.g. from accumulation of marine aerosols or light).

Signal attenuation: We have done tests previously in a lab on a 30 m Teflon tube at similar flow rates; we injected standards of methanol and acetone from the inlet tip and didn't observe any obvious losses down the tube. We hadn't tested other VOCs rigorously. However, given the fact that methanol is the most soluble of the compounds we measure, its negligible loss suggests that the other VOCs we measure should have high transmission through the tube as well.

Signal contamination: We acknowledge that our air sampling tube was longer than what we had hoped for. This was due to logistical constraints on the James Clark Ross, as 90 m is the shortest distance possible from a location of fair low air contamination (i.e. the bridge) and a lab big enough to house the PTR-MS. The sampling tube followed a complex path around the ship, had a number of tight turns, and was mostly sheltered from direct sunlight. We expect that the tight turns had removed much of the larger aerosols from the sampled air, similar to impactors. The main inlet flow was about 30 Lpm and the residence time was fairly short at ∼6s. The PTR-MS subsampled from the main sample flow with only ca. 100 ml/min. We do not expect large aerosols to make it to the PTR-MS because of the tight turns in the main sampling tube
as well as the low subsample flow. The light dependant contamination of acetaldehyde and acetone noted by Yang ACP 2014 on the AMT22 cruise was due to a the usage a plastic funnel on the front of the inlet, which was not used during this deployment.

Finally, we note that several other published works used long Teflon inlet tubes and had even longer air residence times than our setup. For example, Williams et al. (2010) use a 75 m Teflon tube and quote their residence time as less than 1 min for measurements of isoprene. Colomb et al. (2009) used a 80 m Teflon tube with a residence time estimated as less than 2 min for measurements of a large number of OVOCs. Marandino et al. (2005) used a 75 m Teflon tube with a delay time of 12 s for measurements of acetone.

(2) Are the atmospheric concentration units (nmol/L) in Table 1 correct? Elsewhere the air values are quoted as ppbv (nmol/mol, nL/L). Many values reported in the text would fall well below the DL if nmol/L values in Table 1 are converted to ppbv. If air units on Table 1 are in fact ppbv, then project mean concentrations for all species but DMS appear to be right at the DL. I'll assume this is the case, but the authors should clarify.

Indeed, we quoted the wrong units in table 1. The limit of detection and measurement noise of the ambient air measurements are expressed as ppbv, not as nmol dm-3. Thank you for spotting our error.

(3) There are many places in the text where the authors state their measurements 'compare well' with prior studies, but specific values from the literature are not always given. A comparison with published studies in the Southern Ocean (SO) and other regions is important but would be easier to digest if this information were removed from the various results sections, organized, and presented in Table format. A discussion of the these should be provided in a Discussion Section following the Results.

Suggestion accepted. Following reviewer's comments, two tables (Table 5 and Table 6) have been added to the manuscript summarising previous seawater and ambient marine air measurements.

(4) We know more about DMS than the other species in this study, and the surface ocean is unambiguously a source of DMS to the atmosphere over all seasons. Assuming the seawater concentrations and estimated fluxes observed in low-productivity areas are generally representative of fall/winter conditions over the entire SO, and the mean values from the entire cruise are typical of summer, it would be interesting to compute the estimated annual DMS emission over the entire SO region and compare this with prior estimates. Do we think the results from this cruise are representative of the SO in general? Have we now reached a reasonable consensus on annual DMS emissions from the SO?

According to Lana et al. 2011, the uncertainty of the predicted seasonal amplitude in DMS flux at this latitude is at least one order of magnitude. There is also a large spatial variability in DMS flux during the warmer months. Thus it seems risky to assume that our measurements from this cruise (even though it covered a long distance) will be representative of the Southern Ocean. We would be delighted for our data to be incorporated into the generation of the seawater DMS climatology. Pooling all of those data together will help us answer the question of DMS emissions from the Southern Ocean.

(5) This project is valuable because the SO is a unique marine environment, isolated from anthropogenic and continental emission sources. The cruise covered a broad swath of the SO, encountered a range of conditions relating to primary productivity, and conducted the first survey of air/sea concentrations for methanol, acetone and acetaldehyde. Readers will inevitably speculate on the broader geochemical significance of the results, so it seems to me the authors could strengthen their concluding remarks and provide their own perspective, suggesting hypotheses that emerge from this study. For example, from what I've gathered in my brief reading: 1. The SO is supersaturated with isoprene, even in low productivity areas, implying a continuous source to the marine atmosphere, perhaps over all seasons. 2. Methanol, acetone (and acetaldehyde?) are undersaturated in the surface ocean except during episodic

cases of enhanced productivity, and it is therefore likely the undersaturated condition persists throughout the fall/winter seasons when productivity is low. Thus, the SO represents a sink in the global atmospheric budgets for methanol and acetone. 3. The observed relationships to fCO2 provide a crude way to estimate localized emissions of these gas species and their impact on atmospheric oxidative capacity and aerosol production/growth. We hypothesize these atmospheric impacts are restricted to upwelling regions of high productivity. Are these appropriate? I'd like to hear the author's thoughts

1. Suggestion accepted, see below. Isoprene was supersaturated by 760 % in the mean. The large supersaturation and low solubility of isoprene suggest that ambient air mixing ratios influence isoprene saturation levels very little. Our data also shows that isoprene is consistently oversaturated, even in less biologically productive areas, implying a consistent flux out of the ocean, perhaps year round. A mean isoprene flux of 0.028 $\mu$mole m-2 d-1.is computed for this deployment, which exceeded 0.07 $\mu$mole m-2 d-1 on occasions.

2. Suggestion accepted, see below. The high resolution and frequent alternation between ambient air and seawater measurements allowed us to compute the fluxes and saturations for all of these compounds at a high temporal/spatial resolution. This improves the accuracy in the estimated flux since they capture the fine scale variability in the flux direction/magnitude. DMS flux to the atmosphere varied by more than an order of magnitude, with the largest emission associated with a phytoplankton bloom. The Southern Ocean is strongly and consistently supersaturated in isoprene, implying a continuous source of isoprene to the marine atmosphere from the surface ocean, probably year round. Methanol was transferred mostly from the atmosphere to the ocean during this cruise, giving a campaign mean flux of -2.3 $\mu$mol m-2 d-1. However, episodes of high methanol seawater concentrations were observed within a phytoplankton bloom, which led to somewhat unexpected occasions of methanol outgassing from the ocean. Due to the high solubility of methanol and the fact that outgassing was

observed only in very productive areas, we hypothesise that the Southern Ocean is on average a net sink of methanol year round. Acetone and acetaldehyde were both absorbed and emitted by the ocean depending on location. This sector of the Southern Ocean was calculated to be a very weak sink of acetone and acetaldehyde during this period, with a mean flux of -0.55 $\mu$mol m-2 d-1 and -0.24 $\mu$mol m-2 d-1 respectively. Given that these measurements were made in the summer/autumn, when there was still reasonable light and biological activity, it seems unlikely for the Southern Ocean to be a net source of acetone and acetaldehyde when annually averaged.

3. Suggestion accepted, see below Simultaneous measurement of multiple compounds allowed possible common sources and sinks to be identified. For example, seawater methanol and isoprene concentrations were found to positively correlate, possibly due to similar biological sources for these two gases. Isoprene seawater concentrations were found to negatively correlate with fCO2 and with chlorophyll a, supporting a biological origin for isoprene. Seawater acetone and methanol concentrations were found to correlate negatively with fCO2, possibly pointing towards biological sources in seawater. These correlations are perhaps more obvious to the Southern Ocean due to the remoteness and solely marine influence. We suggest that fCO2 may be one of the key factors in predicting seawater isoprene, methanol and acetone in the Southern Ocean. Acetaldehyde concentrations did not clearly correlate with the other gases, possibly due to its strong photochemical production and very rapid oxidation by bacteria (Dixon et al., 2013) which prevented significant accumulations. The observations presented here represent a unique dataset that can be used in models to elucidate more accurately not only the role of the ocean in the cycling of these VOCs, but also the impact of these VOCs on the atmosphere. In particular, elevated concentrations of seawater DMS, isoprene, methanol, and acetone were observed in Southern Ocean phytoplankton blooms. We expect the atmosphere downwind of these hot spots of emission to be the most impacted in terms of atmospheric oxidative capacity, aerosols and clouds.

Minor comments:

line 32: 'Dimethyl sulfide is a key source of secondary organic aerosol' - suggest you omit 'organic' since the major contribution is from inorganic sulfate, although MSA is also produced.

Suggestion accepted.

line 39: I wouldn't call PAN a 'pollutant' since it's a natural component of photochemical cycles in the background (unpolluted) atmosphere.

Suggestion accepted.

line 228: do you mean 'H is the dimensionless liquid over gas' form of H? Suggestion accepted. Thank you for spotting this mistake. Finally, it seems like the Appendix and related plots on solubility could be moved to the supplemental material.

We would prefer to keep this information in the appendix and as part of the main manuscript. This is to increase awareness and credibility of these suggested improved solubilities of methanol and acetone. We believe they are important for a more accurate estimate of emission of these gases from the surface ocean in global models for example.

Please also note the supplement to this comment:
https://www.biogeosciences-discuss.net/bg-2020-2/bg-2020-2-AC1-supplement.pdf

---

## Author Comment (AC2) · 1 Apr 2020

Many thanks for the thoughtful comments from this reviewer. The reviewer has been able to provide thought provoking comments which in our opinion improved the manuscript. Please see our responses below. Reviewer comments are in italic and author's replies can be found in normal font. The changes to the manuscript are presented in red font colour.

The authors mention that, to their knowledge, these are the first reported seawater measurements for methanol, acetone and acetaldehyde. Even for DMS and isoprene the Southern Ocean is highly undersampled which, for all 5 compounds, increases

errors when running global atmospheric models and using no (or very sparse) data from the Southern Ocean. This fact highlights the importance of these measurements presented in the manuscript, which I think the authors are aware of. However, I suggest to even highlight this importance in the introduction section adding a paragraph about the Southern Ocean and its influence on the atmospheric chemistry, highlighting the importance of this work.

Suggestion accepted, please see below.

L74: Models indicate that over the Southern Ocean and globally, DMS (Tesdal et al., 2016) and isoprene (Carslaw et al., 2013) emissions are important for cloud formation and the albedo of the planet. The Southern Ocean is highly under-sampled for DMS and isoprene which increases errors when running global atmospheric models and using no (or very sparse) data from the Southern Ocean. To give an appreciation of the sensitivity of the models to these emissions, Woodhouse et al. (2013) calculate a 4-6 % change in global CCN for a 10 % change in DMS flux (relative to Kettle and Andreae (2000)) in the Atlantic sector of the Southern Ocean for December. Variations in CCN concentrations show clear seasonal trends with highest concentrations typically observed in austral summer (Kim et al., 2017a) thus suggesting, amongst others, a role of biological productivity in formation of CCN over the Southern Ocean.

2.1.1: Calibrations. Do the authors have any scientific explanation why isoprene, the most insoluble compound within the 5 compounds presented, is the only compound not achieving fully equilibration using the presented setup?

The reason for incomplete equilibration of isoprene is discussed in detail in a manuscript describing the technique used to make these measurements (Wohl et al., 2019). To briefly recap, the degree of equilibration depends on solubility of the compound due to the dependance of the air sea exchange onstant on solubility (Liss and Slater, 1974). Less soluble compounds are expected to equilibrate slower.

2.2.3: The authors mention the light driven contamination in the seawater measurements of acetone, acetaldehyde and isoprene and I am confident that they solved this issue. However, for me it is not clear what exactly causes this issue. Perhaps the authors could state clearly if they think it is coming from contamination of the material exposed to high sunlight intensity or from photochemical production in the water flowing through the tube. Both facts seem reasonable, however, if it is the material shouldn't you see these variations also when measuring outside air? Did you experience similar issues in former cruises or tests?

For us it also remains unclear what exactly caused this issue. When sunlight was shining directly at the equilibrator through the window, VOC levels measured with the PTR-MS/SFCE immediately increased to unrealistically high levels. When we closed the blinds, the measured VOC levels greatly decreased and the change was again immediate. A large reduction in the VOC levels was also observed when shielding the air-water separating tee from direct sunlight. The residence time of sample air in this air-water separating tee was on the order of half a minute.

Concerning the outsie air sampling line: the residence time in the air sampling tube was short at approximately 6 s. Most of the 90 m long air inlet tube was also shielded from direct sunlight. We have not experienced issues on former cruises with Teflon air inlets. And our only previous deployment of the SFCE (Wohl et al. 2019) at sea was in a windows-less lab. Photochemical productions of isoprene and carbonyl compounds at the sea surface microlayer has been observed before (Brüggemann et al., 2018; Ciuraru et al., 2015). It's possible that similar reactions were taking place on the water surfaces inside of the SFCE. However such photochemical productions were not evident in the air measurement because the air sampling tube was usually dry and the air residence time was much shorter.

A few sentences have been added to the manuscript:

The exact cause of this light-driven contamination in the SFCE system is unclear. Photochemical production of isoprene and carbonyl compounds at the sea surface microlayer has been observed before (Brüggemann et al., 2018; Ciuraru et al., 2015). It could be that similar reactions were taking place on the water surfaces inside of the SFCE.

Figure 3: Data is shown for 2 weeks before and after dealing with this issue. First, the cruise started only one week before the issue was dealt with, which leads to second, do the two subsets of data have about the same number of measurements? Please check. Additionally, the authors state that daytime values prior to 04/03/19 were not used. However, it seems, that night time data shown in Figure 3 is consistently (for all three compounds) lower than data prior to 04/03/19 shown in Fig. 5a, 7a, and 8a. (i.e. Figure 3, acetone seawater night time values "2 weeks before": âĹij6 nM; Figure 7, average acetone seawater values shown prior to 04/03/19: âĹij7-9nM). Please check.

Indeed, the cruise started one week before the issue was dealt with. This has been corrected in the text of the manuscript. Bespoke analysis was carried out on 5 min averaged data, where each hour of measurements would contain seven 5 min averaged datapoints. The hourly bins before the SFCE was protected from light contain between 8 and 24 5 min averaged datapoints. The hourly bins after the SFCE was protected from light contain between 30 and 81 5 min averaged datapoints. Generally the fewest datapoints were for daytime measurements, due to operations during daytime.

Thank you for pointing out this inconsistency which has been corrected for in the updated manuscript.

l. 309 / Table 2: The authors mention the positive skewness (mean: 0.053, median: 0.045) in the isoprene ambient air mixing ratio which they explain by biology and wind driven emissions as well as the very short atmospheric lifetime. I totally agree. However, this skewness is not at all discussed in the DMS section although the skewness is way higher (mean: 2.6, median: 1.39). DMS has a longer atmospheric lifetime and is more soluble than isoprene.

The reviewer appears to be referring here to ambient air DMS mixing ratios, while citing

values of the DMS seawater concentrations. The DMS ambient air mixing ratios do not display a strong positive skewness. Therefore we assume that the reviewer refers to the DMS seawater concentrations. The suggestion has been accepted and this skewness is discussed in a little bit more detail. See below. The campaign mean seawater concentration of DMS was 2.60 nmol dm-3 and the median was 1.39 nmol dm-3. This illustrates the positive skewness of the DMS seawater concentrations due to episodic high concentrations of DMS.The highest DMS seawater concentrations were observed near the Antarctic Peninsula upwelling region (around 28/02/19, up to 7.55 nmol dm-3) and east of the South Sandwich Islands (around 13/03/19, up to 24.44 nmol dm-3). Chlorophyll a was also elevated in those regions.

Technical corrections: l. 11: delete "and" after "compounds" Suggestion accepted. l. 27: missing full stop after "outgassing" Suggestion accepted. l. 288: "As shown in Fig. 1 and Fig. 2 . . ." Suggestion accepted. 292: "chlorophyll a" Suggestion accepted. l. 441: "dependent" Suggestion accepted. l. 452: remove "in" Suggestion accepted. l. 518: add "," after "isoprene" Suggestion accepted. ll. 726-730: check reference Reference checked and corrected. typo Figure caption 1: double use of "data" Typo corrected. Figure 2a: right y-axis: remove "(PSU)" Suggestion accepted. Figure 2b: left y-axis "$\mu$g dm-3" Suggestion accepted.Figure caption 5: ". . .and time series of chlorophyll a." Typo corrected.

Please also note the supplement to this comment:
https://www.biogeosciences-discuss.net/bg-2020-2/bg-2020-2-AC2-supplement.pdf